# Past water flow beneath Pine Island and Thwaites glaciers, West Antarctica

James D. Kirkham[1,2], Kelly A. Hogan[2], Robert D. Larter[2], Neil S. Arnold[1], Frank O. Nitsche[3], Nicholas R. Golledge[4], Julian A. Dowdeswell[1]

[1]Scott Polar Research Institute, University of Cambridge, Lensfield Road, Cambridge, CB2 1ER, UK.
[2]British Antarctic Survey, High Cross, Madingley Road, Cambridge CB3 0ET, UK.
[3]Lamont-Doherty Earth Observatory of Columbia University, P.O. Box 1000, Palisades, NY 10964-8000 USA.
[4]Antarctic Research Centre, Victoria University of Wellington, Wellington 6140, New Zealand.

*Correspondence to*: jk675@cam.ac.uk

**Abstract**

Outburst floods from subglacial lakes beneath the Antarctic Ice Sheet modulate ice-flow velocities over periods of months to years. Although subglacial lake-drainage events have been observed from satellite-altimetric data, little is known about their role in the long-term evolution of ice-sheet basal hydrology. Here, we systematically map and model past water flow through an extensive area containing over 1000 subglacial channels and 19 former lake basins exposed on over 19,000 km$^2$ of seafloor by the retreat of Pine Island and Thwaites glaciers, West Antarctica. At 507 m wide and 43 m deep on average, the channels offshore of present-day Pine Island and Thwaites glaciers are approximately twice as deep, three times as wide, and cover an area over 400 times larger than the terrestrial meltwater channels comprising the Labyrinth in the Antarctic Dry Valleys. The channels incised into bedrock offshore of contemporary Pine Island and Thwaites glaciers would have been capable of accommodating discharges of up to $8.8 \times 10^6$ m$^3$ s$^{-1}$. We suggest that the channels were formed by episodic discharges from subglacial lakes trapped during ice-sheet advance and retreat over multiple glacial periods. Our results document the widespread influence of episodic subglacial drainage events during past glacial periods, in particular beneath large ice streams similar to those that continue to dominate contemporary ice-sheet discharge.

## 1    Introduction

The widespread and accelerating retreat of Pine Island and Thwaites glaciers constitutes a potential threat to the stability of the West Antarctic Ice Sheet (WAIS) (Rignot *et al*., 2008, 2014; Joughin *et al*., 2014; Feldmann and Levermann, 2015; Shepherd *et al*., 2018; Yu *et al*., 2018). The routing, storage, and expulsion of subglacial water from beneath the WAIS directly influences its mass-loss rates and, accordingly, sea-level rise (Joughin *et al*., 2002; Alley *et al*., 2006; Bell *et al*., 2007; Stearns *et al*., 2008). Variability in subglacial water supply can lead to ice-sheet instability (Schoof, 2010). However, the effect of subglacial water as a lubricant at the basal ice-sheet boundary is still insufficiently understood to be accurately incorporated into the current generation of ice-sheet models, and is thus presently absent from assessments of future ice-sheet behaviour (Fricker and Scambos, 2009; Flowers, 2015, Fricker *et al*., 2016).

Over recent decades, ice-penetrating radio-echo sounding surveys and satellite altimetry have revealed an intricate subglacial network of water storage and transfer beneath the contemporary Antarctic Ice Sheet. Over 400 ponded water bodies, termed subglacial lakes, have been detected beneath the ice (Fig. 1) (e.g. Robin *et al*., 1970; Oswald and

Robin, 1973; Siegert *et al*., 1996, 2005, 2015), about a quarter of which have been observed to fill and drain over sub-decadal timescales (Gray *et al*., 2005; Wingham *et al*., 2006; Fricker *et al*., 2007; Smith *et al*., 2009a, 2017). The subglacial routing of the water released from this special class of 'active' subglacial lakes can be traced for hundreds of kilometres, often triggering a cascade of further subglacial lake drainage downstream (Wingham *et al*., 2006; Flament *et al*., 2014; Fricker *et al*., 2007, 2014). In some instances, active lake drainage has been associated with temporary accelerations in ice velocity (Stearns *et al*., 2008; Siegfried *et al*., 2016). Active Antarctic subglacial lakes typically discharge relatively small volumes of water over several months, with peak discharges of tens of cubic metres per second. For example, lakes detected in the Adventure subglacial trench, East Antarctica, discharged 1.8 km$^3$ of water over 16 months with a peak discharge of ~50 m$^3$ s$^{-1}$ (Wingham *et al*., 2006), whilst 3.7 km$^3$ of water was released from a system of subglacial lakes beneath Thwaites Glacier between June 2013 and January 2014 (Smith *et al*., 2017). To date, the largest contemporary subglacial lake drainage event observed in Antarctica was the loss of $5.2 \pm 1.5$ km$^3$ of water from Lake Cook, East Antarctica, over a 2-year period (Flament *et al*., 2014).

Geomorphological evidence suggests that subglacial water movement, substantially larger than that documented in the satellite era, has occurred beneath the Antarctic Ice Sheet in the past. The mountains flanking the Antarctic Dry Valleys in southern Victoria Land and the ice-free margin of the East Antarctic Soya Coast contain abundant features formed by the flow of subglacial meltwater (Denton *et al*., 1984; Sawagaki and Hirakawa, 1997). Channel systems that are kilometres long and hundreds of metres wide are present in the Asgard Range (Sugden *et al*., 1991), in the foothills of the Royal Society Range (Sugden *et al*., 1999), and in the Convoy Range of the Transantarctic Mountains (Denton and Sugden, 2005; Lewis *et al*., 2006) (Fig. 1). One of the most spectacular relict Antarctic landscapes carved by subglacial water is a 5-km long anastomosing system of terrestrial channels in Wright Valley termed the Labyrinth (Selby and Wilson, 1971). The Labyrinth channels exceed 500 m in width and are incised over 150 m into a 300 m thick sill of Ferrar Dolerite (Lewis *et al*., 2006). Channel formation has been attributed to repeated subglacial outburst floods, with estimated discharges of up to 1.6–2.2×10$^6$ m$^3$ s$^{-1}$, sourced from one or more subglacial lakes that became trapped as the Antarctic Ice Sheet overrode the Transantarctic Mountains during the Miocene epoch (23–5.3 Ma) (Marchant *et al*., 1993, 1996; Denton and Sugden, 2005; Lewis *et al*., 2006). The irregular reverse gradients, anastomosing structure, and the potholes and plunge pools associated with the channel systems in southern Victoria Land are consistent with formation by pressurised subglacial meltwater (Shreve, 1972; Denton and Sugden, 2005; Marchant *et al*., 2011) on a scale that is perhaps unequalled outside the Channeled Scabland landscape of eastern Washington State — produced by some of the largest sub-aerial floods on Earth from Pleistocene proglacial Lake Missoula (Bretz, 1923, 1969).

Channels with similar dimensions have been identified submerged offshore on the largely bedrock-dominated inner continental shelves of the western Antarctic Peninsula (e.g. Ó Cofaigh *et al*., 2002, 2005; Domack *et al*., 2006; Anderson and Oakes-Fretwell, 2008), West Antarctica (e.g. Lowe and Anderson, 2002, 2003; Larter *et al*., 2009; Nitsche *et al*., 2013), and eroded into soft sediments in the inner Ross Sea (Simkins *et al*., 2017). The observed landforms are typically hundreds of metres wide, tens of metres deep, and possess undulating thalwegs that indicate incision by pressurised subglacial meltwater (Lowe and Anderson, 2002). The size and widespread distribution of the offshore channels implies that substantial quantities of subglacial meltwater were required for their formation; this inference is fundamentally inconsistent with the quantity of meltwater produced at the Antarctic ice-bed interface currently (Nitsche *et al*., 2013; Rose *et al*., 2014) and the discharges associated with active subglacial lake water transfer (Wingham *et al*., 2006; Stearns *et al*., 2008). The physical process responsible for the formation of the large channel systems on the

Antarctic continental shelf thus remains unresolved, limiting our understanding of ice-sheet hydrology and future ice-sheet behaviour.

In this study, we examine the origin and formation of a huge system of submarine channels, covering an area >19,000 km$^2$ (over twice the size of Yellowstone National Park), located within 200 km of the present-day margins of Pine Island and Thwaites glaciers. We map the detailed channel network and conduct a quantitative morphometric comparison between the channels in Pine Island Bay and the terrestrial channels comprising the Labyrinth. We then use a numerical model to simulate the subglacial hydrological conditions and water routing beneath Pine Island and Thwaites glaciers during the Last Glacial Maximum (LGM), and provide constraints on the possible mechanisms and timescales over which the submarine channels formed.

## 2        Methods

### 2.1        Bathymetric and terrestrial data

We used multibeam-bathymetric data, collected over two decades by extensive shipborne surveys in Pine Island Bay, to produce a comprehensive high-resolution digital elevation model (DEM) of the seafloor offshore of modern Pine Island and Thwaites glaciers (Table 1). The multibeam-bathymetric data were gridded at a 20-m resolution. A conservative degree of interpolation, applied up to four cell widths away from a data-filled cell, was used to fill small missing gaps in the grids when proximal to existing bathymetry data. This was the highest resolution that could be achieved without requiring interpolation to fill more than 10 % of the study area's grid cells. In addition to the bathymetric data, a 2-m horizontal resolution DEM of Wright Valley, in the McMurdo Dry Valleys of Antarctica, was obtained from the National Aeronautics and Space Administration's 2001 Airborne Topographic Mapper laser altimetry survey to examine the morphology of the Labyrinth (Schenk *et al*., 2004).

### 2.2        Derivation of channel metrics

Channels were digitised manually using hill-shaded DEMs in ArcGIS 10.4. Comparison with a three-dimensional representation of the study area, in which the digitised channel outlines were draped over the projected DEM, ensured that the boundaries of the channels were demarcated accurately (Mayer *et al*., 2000). This technique permitted the routing pathway of each individual channel to be established. Channels were defined to terminate when they were intersected by a larger, deeper routing pathway. We repeated this analysis for the Labyrinth channels which, because of their size and anastomosing structure, are arguably the closest analogue for the channels submerged offshore of Pine Island and Thwaites glaciers.

More than 4200 cross sections were analysed to determine the morphological characteristics of the now-submarine Pine Island and Thwaites glacier channels, whilst more than 1600 were used to analyse the subaerially exposed Labyrinth channels. Based on Noormets *et al*. (2009), we developed a semi-automated algorithm to quantify the width, depth, cross-sectional area, and symmetry of each channel cross-section (Fig. 2). Channel width is defined as the distance between the highest point of each of the channel sides. The greatest vertical distance between the base of the channel and a line intersecting the two channel edges defines the channel depth, *D*. The horizontal distances from the deepest point of the channel to each of the channel sides, *W1* and *W2*, are calculated using an idealised triangular

representation of the channel. The normalised ratio of *W1* to *W2* is used to assess the symmetry of the channel, which can range between 0 and 2. Symmetry values of 1 indicate a symmetrical cross-section, whilst values greater than 1 signify a left-skewed channel and values less than 1 indicate a right-skewed channel cross section. Trapezoidal numerical integration was used to approximate the internal area of the channel, bounded by the line intersecting the channel edges. The ratio of channel depth to channel width was used to enable comparison between the vertical and horizontal proportions of each cross section.

In addition to the channel depth-width ratio, the cross-sectional shape of each sample was assessed using the General Power Law (GPL) program (Pattyn and Van Huele, 1998). The GPL program applies a general form of the power-law equation to derive a best-fit approximation of a shape parameter, *b*, which summarises the geometric shape of the channel:

$$y - y_0 = a \mid x - x_0 \mid^b \tag{1}$$

where *a* and *b* are constants, $x_0$ and $y_0$ are the coordinates of the origin of the cross section and *x* and *y* are the horizontal and vertical components of the channel cross section, respectively. A general least-squares method is used to derive the best-fit values of *a* and *b*. The value of the shape parameter, *b*, generally varies from 1 to 2 according to cross-sectional shape, with *b*-values of 1 indicating a 'V-shaped' cross section and values of 2 denoting a perfectly 'U-shaped' channel geometry (Pattyn and Van Huele, 1998). Composite channel forms are typically associated with *b*-values between these two values. Shape-parameter values exceeding 2 are associated with more box-shaped channel cross-sectional profiles, whereas geometries characterised by *b*-values less than 1 indicate channel flanks with a convex-upward form (Gales *et al*., 2013).

### 2.3    Modelling past subglacial water flow

To assess the rates at which subglacial water could be produced, stored and routed beneath the former Pine Island and Thwaites glaciers, we performed hydrological modelling simulations for the full-glacial configuration of the Pine Island/Thwaites catchment, accounting for isostatic loading under the expanded WAIS, at 20 ka.

### 2.3.1    Palaeo ice-sheet reconstruction and data

The thickness and surface slope of the LGM configuration of the WAIS was derived from a series of palaeo ice-sheet reconstructions produced using the Parallel Ice Sheet Model (PISM) (Golledge *et al*., 2012). PISM is a three-dimensional, thermomechanical ice-sheet model constrained by geological observations that has been widely used to simulate the dynamics of the Antarctic ice sheets (e.g. Golledge *et al*., 2012, 2013, 2014; Fogwill *et al*., 2014). PISM combines shallow-ice and shallow-shelf approximations for grounded ice to capture the dynamic behaviour of grounded ice and is able to simulate ice-stream flow and ice drawdown at a relatively high 5-km resolution (Golledge *et al*., 2012; Fogwill *et al*., 2014).

Simulation results from Golledge *et al*. (2012) provided the data required to calculate the routing and flux of subglacial water flowing under hydrostatic pressure beneath Pine Island and Thwaites glaciers at the LGM. A nested modelling approach was adopted in which catchment-wide water fluxes were first calculated at a 500-m resolution. These model results were then input at the edges of the higher-resolution bathymetric DEM covering the offshore channels. The

bathymetric DEM was resampled to 90 m for the sake of computational efficiency. Ice-surface topography at the LGM was interpolated from the Golledge *et al.* (2012) reconstructions to the appropriate resolution (500 m or 90 m). Bed topography for the majority of the catchment was derived from the 500-m resolution BedMachine Antarctica dataset of the contemporary Antarctic Ice Sheet's bed (Millan *et al.*, 2017; Morlighem *et al.*, 2018). These data were corrected for isostasy by calculating the isostatic deflection at the LGM from the Golledge *et al.* (2012) results, and then adding the interpolated deflection values to the bed-topography data to produce a 500-m resolution, isostatically corrected DEM. Ice thickness was calculated by subtracting the interpolated ice-sheet surface from the isostatically-corrected bed topography. Subglacial topography and ice thickness were then used to calculate the subglacial hydraulic potential within the study region.

Basal meltwater production was calculated using the modelled basal frictional heating from the Golledge *et al.* (2012) reconstruction, added to the geothermal heat flux of the LGM Pine Island and Thwaites glaciers catchment. Geothermal heat flux values were derived from Martos *et al.* (2017) for the portion covered by the contemporary ice sheet and from Davies (2013) for the offshore portion of formerly ice-covered land. The geothermal heat flux of the catchment ranges between ~70 mWm$^{-2}$ and ~130 mWm$^{-2}$, whilst frictional heating values range from ~0 mWm$^{-2}$ to ~500 mWm$^{-2}$. For areas of the bed calculated to be at the pressure melting point in the Golledge *et al.* (2012) results, the sum of these heat sources was used to calculate the basal melt rate in each DEM cell, which can then be multiplied by the cell area to give a water volume per unit time. For cold-based ice where basal meltwater is absent, the production of basal meltwater was set to zero.

### 2.3.2    Flow routing

Meltwater routing beneath the ~600,000 km$^2$ area drained by the LGM Pine Island and Thwaites glaciers catchment was modelled using a weighted upstream catchment area algorithm (Arnold, 2010; Willis *et al.* 2016). Subglacial hydraulic potential, Φ, dictates the flow and routing of water beneath ice masses (Shreve, 1972). The subglacial hydraulic potential of water flowing beneath an ice sheet is a function of bed topography and ice thickness:

$$\Phi = \rho_w g h + k\rho_i g Z \tag{2}$$

where $\rho_w$ and $\rho_i$ are the densities of water and ice (kg m$^{-3}$) respectively, $g$ is the acceleration due to gravity (m s$^{-2}$), $h$ is bed elevation (m), $Z$ is ice thickness (m) and $k$ is a dimensionless parameter (referred to as the uniform floatation factor) that can be varied from 0 to 1 to simulate variations in subglacial water pressure. Higher $k$ values represent greater subglacial water pressure, with $k = 1$ denoting that water pressure is equal to the overburden pressure exerted by the overlying ice, and $k = 0$ representing water flowing at atmospheric pressure. Seismic (Blankenship *et al.*, 1986; Alley *et al.*, 1986) and borehole (Engelhardt *et al.*, 1990; Kamb, 2001; Tulaczyk *et al.*, 2001) investigations of the basal properties of Antarctic ice streams demonstrate that, where present, the pressure of subglacial meltwater is typically close to the ice-overburden pressure. Multibeam swath bathymetry, sub-bottom profiler, and coring studies have previously established that the now-submarine study area in Pine Island Bay was occupied by a major ice stream during the Last Glacial Maximum (e.g. Graham *et al.*, 2010). A $k$ value of 0.995 was therefore chosen to parameterise subglacial water pressures within the numerical-model simulations to approximate the subglacial water pressures observed under contemporary Antarctic ice streams.

The upstream area model is described fully in Arnold (2010), and its adaptation for calculating the volume, throughput water discharge, and possible residence time of subglacial lakes is described in Willis *et al.* (2016). Briefly, the algorithm first identifies all cells in a hydraulic potential surface that have a lower potential than their neighbours. Such 'sink' cells allow local subglacial catchments (a group of contiguous cells which all drain toward the sink) to be determined, and also form the nucleus for potential subglacial lakes. The algorithm 'floods' each sink cell to find the elevation of the lowest cell in the catchment surrounding the sink over which water would spill into a lower potential downstream cell (and hence, into an adjacent catchment). This spill point cell defines the maximum depth (relative to the sink), area and volume of each potential lake, and also allows the routing algorithm to pass the total catchment area (or cumulative subglacial melt) from catchment to catchment downstream until the model reaches the edge of the DEM. In this way, the algorithm builds up the topology of subglacial water flow, linking the individual catchments together into arborescent structures. The algorithm thus calculates the volume and direction of subglacial water flow; however, it is not capable of predicting physical flow conditions within individual channels.

Rather than simply accumulating the DEM cell areas, the algorithm is weighted such that each cell contributes its modelled basal melt flux (see Section 2.3.1). This enables the steady-state subglacial water flux in each cell within the DEM, and the total discharge flowing into each potential subglacial lake, to be calculated. For each lake, the outflow discharge into the adjacent downstream catchment is set to be equal to the inflow discharge, as the algorithm assumes steady-state flow through each lake (which is assumed to be at maximum volume) in order to maintain topological continuity. By dividing the volume of each lake by the input water flux, the total time to refill the lake (again assuming steady flow) can be calculated. This effectively gives the minimum duration of any possible fill/drain cycle for each lake.

We compare our modelled continuous subglacial water fluxes to the theoretical discharge that could be accommodated by the channels if sufficient water was available based on channel cross-sectional area (Walder, 1986; Wingham *et al.*, 2006; Jordan *et al.*, 2010). The discharge, $Q$, of a semi-circular cross section, $S$, is equal to:

$$Q = 2 \left(\frac{\pi}{2}\right)^{\frac{1}{3}} S^{\frac{4}{3}} m^{-1} \left(\frac{\emptyset'}{p_w g}\right)^{\frac{1}{2}} \tag{3}$$

where $\emptyset'$ is the hydraulic potential gradient, $p_w$ is the density of water (1000 kg m$^{-3}$), $g$ is acceleration due to gravity (9.81 m s$^{-2}$), and $m$ is the Manning coefficient (0.08 m$^{-1/3}$ s). We derive $S$ from the measured cross-sectional area of the channels in the multibeam bathymetry data and estimate along-channel estimates of $\emptyset'$ using Eq. (2). As the channels were unlikely to have been completely ice-free during periods of occupation by subglacial water, we calculate the hypothetical discharge under three scenarios in which the channel cross sections are: (1) ice-free, (2) 50 % occupied by ice, (3) 90 % occupied by ice.

## 3    Results

### 3.1    Morphology of channels in Pine Island Bay

Over 1000 channels were mapped within Pine Island Bay, covering an area of ~19,000 km$^2$ (Fig. 3). The channels are arranged in a complex, anastomosing pattern that often appears to follow lines of geological weakness in the inner-shelf bedrock. No channels are visible beyond the transition from the inner-shelf bedrock to the sedimentary strata

on the outer continental shelf (e.g. Lowe and Anderson, 2003; Graham *et al*., 2010; Gohl *et al*., 2012; Nitsche *et al*., 2013), ~200 km from the present ice margin. The channels are characterised by undulating long-axis profiles (thalwegs) containing multiple reverse gradients along their lengths. The majority of channels are less than 5000 m long, range in width from 80 m to 3600 m, and are incised into bedrock by between 3 m and 330 m. The channels are 507 m wide and 43 m deep on average, with a typical cross-sectional area of 17,000 $m^2$ (Fig. 4).

The bedrock channel system in Pine Island Bay is interspersed with a series of 19 flat-bottomed depressions with steep sides relative to the gradient of their central floor. The depressions resemble a series of basins connected, and sometimes cross-cut, by the channels. The basins range in planimetric area between 5 $km^2$ and 159 $km^2$ and descend several hundred metres beneath the average depth of the surrounding topography. The basin floors contain subdued glacial landforms (Nitsche *et al*., 2013) that appear to be partially buried by sedimentary infill.

## 3.2    Comparison with the Labyrinth

The channels offshore of present Pine Island and Thwaites glaciers are substantially larger than those comprising the Labyrinth, which consists of a series of 80 channels that are generally less than 1000 m in length, 20 m to 600 m wide, and 2 m to 150 m deep (Fig. 4). The mean cross-sectional area of the Labyrinth channels is 3000 $m^2$. On average, the Labyrinth channels are about half as deep and a third as wide as the Pine Island Bay channels at 23 m and 160 m, respectively. The two channel inventories are comparable in terms of their channel density, sinuosity, and characteristic undulating longitudinal channel profiles; however, the channelised region of Pine Island Bay covers an area more than 400 times larger than the size of the Labyrinth (Figs. 3a–3d).

Both the Labyrinth and the Pine Island Bay channels tend to have asymmetric V-shaped (*b*-value = 1), rather than U-shaped (*b*-value = 2), cross-sections. The base of the cross sections are commonly more V-shaped than the upper sections of the channels. Channel cross-section depth-to-width ratios are also comparable in both regions, with the Labyrinth channels typically 5–12 times as wide as they are deep, whilst the channels in Pine Island Bay are slightly wider in relation to their depth, at 7–25 times as wide as they are deep (Fig. 4). For the Labyrinth channels, increasing cross-sectional area is more strongly correlated with channel depth than channel width, suggesting that channel enlargement predominantly results from disproportionate over-deepening of the larger channels with respect to their width. In contrast, increases in Pine Island Bay channel area are due to both widening and deepening of channel cross sections.

## 3.3    Modelled subglacial hydrology

The numerical model results reveal an intricate system of water transfer and storage beneath the LGM configuration of Pine Island and Thwaites glaciers (Fig. 5). The majority of the LGM catchment is estimated to be at the pressure melting point. Predicted basal melt rates, which are dependent on ice-sheet thickness as well as geothermal and strain heating, are greatest along the thick central and tributary trunks of the expanded Pine Island/Thwaites glaciers, reaching values >50 mm $yr^{-1}$. The basin-wide average melt rate for the LGM catchment is ~20 mm $yr^{-1}$, producing a total meltwater volume of 12.2 $km^3$ per year. The hydraulic potential gradient of the LGM WAIS forces water northwards, forming an arborescent drainage structure that converges into the main trunk of the formerly expanded Pine Island and Thwaites glaciers. Numerous ponded water bodies are predicted to occur within the LGM catchment, several of which

fall within areas where subglacial water has been observed to transfer between subglacial lakes beneath the contemporary WAIS (Smith *et al.*, 2017) (Fig. 5b).

The predicted subglacial water routing converges from across the catchment into the channelised region of Pine Island Bay covered by high-resolution multibeam bathymetry (Fig. 5c). The majority of this water flow is routed preferentially from beneath the present-day glacier margins and into the inner-shelf trough through the bedrock channels (Fig. 5d). The majority of channels have a modelled steady state water discharge of less than 20 $m^3$ $s^{-1}$. The highest calculated continuous meltwater flux occurs in a channel situated in the centre of the trough with a discharge of 139 $m^3$ $s^{-1}$ (Fig. 5c, location (i)). Channels that are wide, deep, and have a large cross-sectional area are generally associated with higher steady-state discharges; these tend to occur in the centre of the trough occupied by the former Pine Island-Thwaites Ice Stream during the LGM (Fig. 6a). Similarly, smaller channels located perpendicular to the former ice stream margins are associated with lower steady-state discharges. No relationship is present between channel symmetry or *b*-value and modelled steady-state water discharge. The largest channels occur in close proximity to the basins in which water is predicted to pond (Fig. 6b). However, although larger channels are associated with higher discharges, this does not hold true in all cases as some of the smallest channels (cross-sectional area <5000 $m^2$) are calculated to carry some of the highest discharges (~130 $m^3$ $s^{-1}$), whilst many of the largest channels contain discharges <0.1 $m^3$ $s^{-1}$ (Fig. 6c). This discrepancy is emphasised at location (i). Here, four tributary channels flow into a basin which then outflows into one large channel that has the highest calculated water flux in the catchment (Fig. 7a). The four tributary channels have approximately the same cross-sectional area as the main outflow channel but have steady-state discharges over three orders of magnitude less than those of the main channel, demonstrating that not all large channels are associated with high steady-state discharges.

Multiple lakes are predicted to occur in the bedrock depressions in Pine Island Bay under LGM conditions. Once the lakes are filled to their spill point, water is transferred through the bedrock channels into further lakes downstream. We examine the modelled fluxes, volumes, and recharge rates of the lakes using four examples displayed in Fig. 5c. Lake (i) occurs at the confluence of the main streams that drain the Pine Island and Thwaites glaciers catchments; lake (ii) is the furthest downstream lake in the Thwaites catchment, and lake (iii) is the furthest downstream lake in the Pine Island catchment. The fourth example lake occurs within the relict subglacial lake basin described by Kuhn *et al.* (2017). The four example lakes have volumes between 0.33 $km^3$ and 19.9 $km^3$ when filled to their spill points. The majority of the water routed into Pine Island Bay is directed from beneath Thwaites Glacier, flowing into lake (ii) at a steady-state rate of 101 $m^3$ $s^{-1}$. When combined with the steady-state flux of water contributed by the Pine Island Glacier catchment (32.2 $m^3$ $s^{-1}$), the two catchments yield a steady-state discharge of 139 $m^3$ $s^{-1}$ through lake (i). Under these continuous flow rates, the three lakes would fill to their spill points on annual to decadal timescales (Table 2). The basin described by Kuhn *et al.* (2017) is smaller than the lakes draining the main glacier catchments and is modelled to accumulate water at a steady state rate of 0.23 $m^3$ $s^{-1}$, filling to its spill point every 45 years. If these example lakes were to drain over a ~16 month period, typical of subglacial water transfer beneath the contemporary Antarctic Ice Sheet (Wingham *et al.*, 2006), water would be released from the lakes at an average rate of ~10–80 $m^3$ $s^{-1}$ for the three smaller lakes (i, ii, iv) and ~470 $m^3$ $s^{-1}$ for lake (iii) at the edge of the individual Pine Island glacier catchment (Table 2).

We quantify the largest possible discharge that could occur through the channels by considering a scenario in which a large modelled lake, situated upstream in the Pine Island/Thwaites catchment, drains within 16 months and triggers a cascade of catastrophic lake drainages downstream. This scenario represents the most extreme discharge that

could have been generated beneath the LGM configuration of Pine Island and Thwaites glaciers. Under this scenario, the mean flood discharge through the inner shelf channels would be ~5.5–6.6 × $10^4$ $m^3$ $s^{-1}$. However, these are mean discharges. In a modelling study of water transfer between subglacial lakes in the Adventure subglacial trench, using data from Wingham *et al*. (2006), Peters *et al*. (2009) found that subglacial discharge between the lakes was highly sensitive to the model parameters used, especially the assumed channel roughness, and short-lived peak fluxes of around ten times the mean discharge were quite conceivable. Given this uncertainty, it may have been possible to produce short-lived outbursts up to ~5 × $10^5$ $m^3$ $s^{-1}$ through the Pine Island Bay channels. This estimate is of the order of 10 % to 50 % of the maximum carrying capacity of the channels (Eq. (3)). Under the hydraulic potential gradient of the LGM configuration of the WAIS, channels filled with water to the bankfull stage would be capable of accommodating flows with an average discharge of ~8.8 × $10^6$ $m^3$ $s^{-1}$. This carrying capacity would be reduced to 3.5 × $10^6$ $m^3$ $s^{-1}$ if the channels were 50 % ice filled, and to 2.8 × $10^5$ $m^3$ $s^{-1}$ if they were 90 % ice filled.

## 5        Discussion

### 5.1        Channel formation

The morphology, discharge carrying capacity, and form distributions of the Pine Island Bay and Labyrinth channels are strikingly similar despite the former being significantly larger. Channel form ratios and *b*-values demonstrate that both sets of channels tend to have broad and shallow V-shaped cross-sectional profiles (Figure 4), indicative of subglacial meltwater erosion, rather than the U-shaped morphology associated with direct glacial erosion (Pattyn and Van Huele, 1998; Rose *et al*., 2014). Some of the channels also have trapezoidal cross-sectional forms (*b*-values ~1.3-1.6; Figure 7c). This cross-sectional shape has been associated with high-magnitude discharges of water (e.g. Bretz, 1923, 1969; Gupta *et al*., 2007; Larsen and Lamb, 2016), although the shape of the channels in Pine Island Bay, and accordingly their *b*-values, may be unrepresentative where significant sediment infill of the base of the channel is present (e.g. Smith *et al*., 2009). The undulating long-axis profiles and reverse gradients associated with both sets of channel inventories, combined with their size, shape and their incision into bedrock suggests that they were formed by high-velocity subglacial meltwater flowing under hydrostatic pressure (Shreve, 1972; Sugden *et al*., 1991; Lowe and Anderson, 2003; Lewis *et al*., 2006; Smith *et al*., 2009b; van der Vegt *et al*., 2012; Nitsche *et al*., 2013). Morphometric evidence thus indicates a similar formative process for both sets of channels; that is, incision by pressurised subglacial water, albeit executed over a vastly larger scale beneath Pine Island and Thwaites glaciers than at the Labyrinth (Figs. 3, 7b, 7c).

The dimensions and the extensive area over which the submarine channels are observed indicates that the hydrological system beneath the formerly expanded Pine Island and Thwaites glaciers at times transported substantial volumes of pressurised water along consistent routing pathways. The inner shelf substrate in Pine Island Bay consists mainly of hard granitoid bedrock and porphyritic dykes (e.g. Pankhurst *et al*., 1993; Kipf *et al*., 2012; Gohl *et al*., 2012; Lindow *et al*., 2016) that would be resistant to erosion by subglacial meltwater. The transportation of coarse bedload permits water to rapidly incise bedrock (Cook *et al*., 2013a). In order to mobilise such suitable sediment loads, large fluxes of water would have been required to excavate channels of the size observed in Pine Island Bay (Alley *et al*., 1997). Although it is likely that infilling with ice prevented all of the channels from being active contemporaneously (Nitsche *et al*., 2013), even if half of the channel cross sections were filled with grounded ice, at 3.5 × $10^6$ $m^3$ $s^{-1}$, their carrying capacity would still have been larger than the maximum bankfull discharge estimated for the Labyrinth (1.6–2.2×$10^6$ $m^3$ $s^{-1}$) (Lewis *et al*., 2006). Constraining the age of the submarine channels, and determining whether any

resculpting by ice occurs within a glacial cycle or over shorter timescales, is difficult. However, the presence of deformation till in some of the channels indicates that they have been overridden by wet-based ice since their incision so may be the outcome of several advances and retreats of grounded ice through Pine Island Bay (Smith *et al*., 2009b; Nitsche *et al*., 2013).

Unlike the channels in Pine Island Bay, the age and formation of the Labyrinth is better constrained, and has been attributed to one or more subglacial floods sourced from a subglacial lake trapped as the Antarctic Ice Sheet overrode the Transantarctic Mountains during the Miocene (Lewis *et al*., 2006). The last period of Labyrinth channel incision occurred between 14.4 Ma and 12.4 Ma (Marchant *et al*., 1993, 1996; Denton and Sugden, 2005; Lewis *et al*., 2006). Around this time, the region experienced strong climatic cooling of at least 8°C (Lewis *et al*., 2007, 2008). This period of cooling changed the basal thermal regime of the ice in the vicinity of the Labyrinth to minimally erosive cold-based ice, potentially protecting the channels from any substantial further erosion (Atkins and Dickinson, 2007). The absence of post-incisional reworking of the top of the channels by wet-based ice may explain the tendency for the Labyrinth channels to be less wide in relation to their depth than the channels in Pine Island Bay (Fig. 4), and for increased Labyrinth channel cross-sectional area to be more strongly correlated with channel depth than channel width. In contrast, increasing cross-sectional area of the channels in Pine Island Bay is correlated with both channel width and depth, suggesting that glacial erosion of the sides of the channels has been a significant additional process influencing the morphology of the channels. This supports the notion that the channels have been formed and reoccupied over multiple glacial cycles, allowing subglacial erosion to enlarge the top of the channels and produce composite features (Fig. 8). Interestingly, this implies that the Labyrinth channels may represent 'purer' meltwater signatures than the larger features observed in Pine Island Bay.

## 5.2     Channel water sources

The polar desert climate characterising the contemporary Antarctic Ice Sheet largely confines the influence of surface melting to the Antarctic Peninsula and the ice shelves fringing the continent (e.g. Tedesco, 2009; Tedesco and Monaghan, 2009; Barrand *et al*., 2013; Trusel *et al*., 2013; Bell *et al*., 2017, 2018). Where present, surface meltwater can be transported efficiently across the ice surface through the action of surficial meltwater rivers (Bell *et al*., 2017; Kingslake *et al*., 2017) or even enter the englacial system and be stored as shallow sub-surface lakes in some ice shelves (Lenaerts *et al*., 2016). However, the absence of surface melting for the majority of the grounded ice sheet results in the water present within the Antarctic subglacial hydrological system being sourced from processes that operate almost exclusively at the ice-sheet bed (Rose *et al*., 2014). The undulating long-profiles of the channels (Figure 3f) indicate that they were formed beneath grounded ice and not from water that propagated through an ice shelf (Nitsche *et al*., 2013). Thus, the hydrological conditions characterising the grounded interior of the contemporary ice sheet offers the best analogue for former subglacial water generation. With average summer temperatures around −10°C at the present-day coast around Pine Island Bay (King and Turner, 2007), the contribution of surface meltwater to the subglacial hydrological system can be considered negligible if the past climate were similar to present, or more likely colder, under late Pleistocene full-glacial conditions (Trusel *et al*., 2013). Therefore, even assuming a climate similar to present, the subglacial meltwater responsible for bedrock-channel formation could only have been generated by geothermal and strain heating at the ice-sheet bed.

The geothermal heat flux beneath West Antarctica is larger than for the remainder of the continent due to its location on the West Antarctic Rift System — an extensional volcanic rift system that stretches across Marie Byrd Land from Pine Island Glacier and into the Ross Sea (Blankenship *et al.*, 1993; LaMasurier, 2013; Loose *et al.*, 2018). Despite this elevated geothermal heat flux, the quantity of subglacial meltwater produced by geothermal heating is estimated to be only 4–10 mm yr$^{-1}$ (Dowdeswell *et al.*, 2016) — two orders of magnitude less than the maximum rates of surface melt observed on the Antarctic Peninsula today (>400 mm yr$^{-1}$ w.e.) (Trusel *et al.*, 2013). Although basal melt rates may be elevated by strain heating at the lateral shear-margins of ice streams (Bell, 2008; Graham *et al.*, 2009), the combination of these processes would only yield subglacial melt rates of up to 90–180 mm yr$^{-1}$ (Dowdeswell *et al.*, 2016).

The basin-wide average basal meltwater production rate beneath Pine Island and Thwaites glaciers is ~28 mm yr$^{-1}$ at present, generating 5.2 km$^3$ of meltwater per year (Joughin *et al.*, 2009). Our model calculates that under the LGM configuration of these glaciers, meltwater would have been produced at ~20 mm yr$^{-1}$ on average, although elevated melt rates >50 mm yr$^{-1}$ would have occurred beneath the thick central ice-stream tongue. Combined, these melt rates would have generated a total volume of 12.2 km$^3$ per year across the entire LGM catchment. These rates are too low to transport sediment efficiently and would therefore preclude significant erosion of bedrock (Alley *et al.*, 1997). This interpretation is supported by the relationship between channel cross-sectional area and modelled steady-state discharge as, if the channels were incised by the continuous flow of meltwater beneath the former Pine Island-Thwaites Ice Stream, we would expect all large channels to be associated with a high discharge whereas this is not the case for all channels (e.g. Figs. 7a, 7c). Furthermore, there is a discrepancy between the potential carrying capacity of the channels and the predicted discharges through the channels if the subglacial water-transfer system is assumed to be in steady state. This discrepancy occurs because the channels, even if 90 % full of ice, would be capable of accommodating discharges over three orders of magnitude larger ($2.8 \times 10^5$ m$^3$ s$^{-1}$) than the largest steady state water fluxes predicted to occur by the numerical model (139 m$^3$ s$^{-1}$). The continuous production of basal meltwater beneath former Pine Island and Thwaites glaciers is therefore insufficient to incise channels of the size present in Pine Island Bay (Lowe and Anderson, 2003; Nitsche *et al.*, 2013). Consequently, another mechanism, capable of mobilising coarse bedload, is required to explain their formation.

Episodic, but high-magnitude, subglacial volcanic activity occurring over multi-millennial timescales may have supplied large volumes of meltwater to the subglacial hydrological system of Pine Island and Thwaites glaciers in the past (Wilch *et al.*, 1999; Nitsche *et al.*, 2013). West Antarctica contains 138 volcanoes, including three extensively eroded Miocene volcanoes and other younger parasitic cones in the Hudson Mountains, 150 km east of Pine Island Bay (Nitsche *et al.*, 2013; Van Wyk de Vries *et al.*, 2017; Loose *et al.*, 2018). The most recent volcanic eruption in this region occurred ~2200 years ago (Corr and Vaughan, 2008). Analogies from subglacial volcanic eruptions in Iceland demonstrate that these events commonly cause meltwater to accumulate in unstable subglacial lakes incised upwards into the ice (Björnsson, 1992, 2002). When hydrostatic pressure exceeds the strength of the ice damming the lake, jökulhlaup outburst floods may occur with observed discharges of up to $5 \times 10^4$ m$^3$ s$^{-1}$ (Björnsson, 2002). Jökulhlaups transport huge amounts of sediment, in the order of 10,000,000 tonnes, per event (Nye, 1976; Roberts, 2005) allowing them to impose a significant geomorphic imprint upon the landscape, including the incision of large subglacial channels (Björnsson, 2002; Russell *et al.*, 2007). However, although it is possible that volcanically induced subglacial floods supplied large volumes of water to the beds of Pine Island and Thwaites glaciers in the past, this mechanism does not explain the occurrence of channelised landforms observed in different, less volcanically active, regions of Antarctica such as the East Antarctic Soya Coast (Sawagaki and Hirakawa, 1997) and the western Antarctic Peninsula (e.g. Ó Cofaigh *et al.*, 2002, 2005; Domack *et al.*,

2006; Anderson and Oakes-Fretwell, 2008). Hence, although episodic volcanism may have contributed water to the formation of the channel system present in Pine Island Bay, it is unlikely to represent the sole or dominant mechanism through which the channels formed.

Meltwater features, similar to the channels in Pine Island Bay, have been mapped and modelled beneath other former ice sheets such as the Fennoscandian and Barents Sea ice sheets (Greenwood *et al*., 2016, 2017; Bjarnadóttir *et al*., 2017; Shackleton *et al*., 2018), and the Laurentide Ice Sheet (Mullins and Hinchey, 1989; Kor *et al*., 1991; Brennand and Shaw, 1994; Livingstone *et al*., 2013a, 2016; Livingstone and Clark, 2016). The deglacial configuration of these former ice sheets was likely conducive to producing large volumes of surface meltwater that may have propagated to the bed (Carlson *et al*., 2008; Jansen *et al*., 2014), similar to that observed on the margins of the contemporary Greenland Ice Sheet (Zwally *et al*., 2002; Das *et al*., 2008; Bartholomew *et al*., 2012). Subglacial streams fed by surface melt have high discharges and an exceptional sediment transport capacity due to the short time period in which supraglacial drainage occurs (Alley *et al*., 1997). When this water is delivered to portions of the bed where unconsolidated sediments have been stripped away, these high water-fluxes will have very large unsatisfied transport capacities and will be able to rapidly erode bedrock if paired with a suitable bedload (Alley *et al*., 1997; Cook *et al*., 2013a). Exceptionally high erosion rates have been associated with supraglacial water input to the Greenland subglacial hydrological system (4.8 ± 2.6 mm a$^{-1}$; 1–2 orders of magnitude larger than long-term estimates of denudation rates beneath the Greenland Ice Sheet) (Andrews *et al*., 1994; Cowton *et al*., 2012), and the input of supraglacial meltwater into the subglacial hydrological system of the deglaciating Fennoscandian Ice Sheet has been proposed to have cut large undulating gorges into bedrock in northern Sweden (Jansen *et al*., 2014).

Although the injection of surface meltwater to the bed of the Antarctic Ice Sheet through hydrofracture is potentially forecast under future climate warming scenarios (Bell *et al*., 2018), the extent to which surface melting of the Antarctic Ice Sheet may have occurred in the past is poorly constrained. However, evidence from environmental proxies (Escutia *et al*., 2009; Cook *et al*., 2013b), sea-level reconstructions (Miller *et al*., 2012) and numerical-modelling simulations (DeConto and Pollard, 2003) indicates that significant circum-Antarctic warming occurred during the Miocene and the Pliocene (5.3–2.6 Ma) epochs. This climate warming may have raised surface temperatures sufficiently to facilitate surface melt across significant portions of the Antarctic interior (Rose *et al*., 2014). However, numerical-model simulations suggest that despite peak Miocene air temperatures being sufficiently warm to facilitate surface melting, it is unlikely that the WAIS was developed enough at this time to allow large-scale grounded ice to extend onto the inner continental shelf of the Amundsen Sea, precluding the possibility of subglacial channel formation in Pine Island Bay (DeConto and Pollard, 2003).

The establishment of a more expansive, yet dynamic (Cook *et al*., 2013b), WAIS during the Pliocene (Pollard and DeConto, 2009; DeConto and Pollard, 2016), combined with temperatures 2°C to 3°C warmer than present (Dowsett, 2007), make this epoch a potential candidate for when substantial quantities of surficial meltwater could propagate to the bed of the WAIS and form the meltwater channels. Ice-sheet modelling by Raymo *et al*. (2006) predicts that the extent of Pliocene summer melting on the East Antarctic Ice Sheet margin would have been comparable to the considerable quantities of supraglacial meltwater produced during the summer ablation season of the contemporary Greenland Ice Sheet (e.g. Abdalati and Steffen, 2001; Zwally *et al*., 2002; Hanna *et al*., 2008). If a comparable level of surface melt also occurred on the surface of the WAIS at this time, substantial quantities of supraglacial meltwater could have propagated to the expanded ice sheet's bed, facilitating channel incision (Rose *et al*., 2014). Although bedrock channels exist on the

Greenland continental shelf (Ó Cofaigh *et al.*, 2004; Dowdeswell *et al.*, 2010) and beneath the contemporary Greenland Ice Sheet (Bamber *et al.*, 2013; Livingstone *et al.*, 2017), they are not nearly as common as those offshore of the WAIS and the formation of many has been attributed to turbidity currents or fluvial activity predating the inception of the Greenland Ice Sheet (e.g. Cooper *et al.*, 2016; Batchelor *et al.*, 2018). Accordingly, if supraglacial water injection is responsible for the incision of the Antarctic channels, it raises the question of why these features are not more common on the margin of the Greenland Ice Sheet where this process is known to occur, or why channels have not been observed forming beneath the contemporary ice sheet today.

An alternative source of water with which the channels could have formed is subglacial lakes produced by the accumulation of subglacial meltwater in hydraulic potential lows. The seafloor of Pine Island Bay and the bed of the LGM catchment contains multiple bedrock basins where low hydraulic potential provides favourable locations for water to pond beneath the WAIS at the LGM (Figs. 5, 7d). The dimensions of the basins beyond the current Pine Island/Thwaites glacier margins are of the same order of magnitude as many contemporary subglacial lakes occupying bedrock basins. For example, Lake Ellsworth has an area of 29 km$^2$ (Siegert *et al.*, 2004) whereas the basins beyond PIG and TG have planimetric areas between 5 km$^2$ and 160 km$^2$. The modelled water bodies have volumes, filling/draining times, and steady state inflow and outflow fluxes that are comparable to contemporary subglacial lake drainage events observed using satellite altimetry (Wingham *et al.*, 2006; Fricker *et al.*, 2007; Flament *et al.*, 2014; Smith *et al.*, 2017). The bedrock basins in Pine Island Bay are infilled with several to tens of metres of sediment (Nitsche *et al.*, 2013). Sediments recovered from one of these basins suggest deposition in a low energy freshwater subglacial lake environment during the last glacial period (Kuhn *et al.*, 2017), an interpretation that is consistent with the low magnitude water flux modelled for the basin sampled (0.23 m$^3$ s$^{-1}$). The fact that the largest channels are located at the edge of the subglacial lakes (Fig. 6b) strongly supports the notion that channel genesis is connected to the basins, possibly through episodic outbursts of water from former subglacial lakes. Consequently, similar to other areas offshore of the Antarctic Peninsula (Domack *et al.*, 2006), it is possible that the bedrock basins in Pine Island Bay may represent the locations of relict subglacial lakes in past glacial periods that episodically drained to form the subglacial channels (Domack *et al.*, 2006; Livingstone *et al.*, 2013b; Kuhn *et al.*, 2017).

Although multiple subglacial lake drainage events have been observed beneath the contemporary Antarctic Ice Sheet (e.g. Gray *et al.*, 2005; Wingham *et al.*, 2006; Fricker *et al.*, 2007; Smith *et al.*, 2009a, 2017), the peak discharge and volume of water displaced during contemporary lake drainage is four to five orders of magnitude smaller than the amount that we predict the channels in Pine Island Bay could accommodate, even when mostly filled with ice. Furthermore, models of active subglacial lake drainage most closely replicate observed subglacial lake behaviour when water mechanically erodes shallow canals into underlying deformable sediment rather than producing an R-channel (Röthlisberger, 1972) type drainage mechanism (Fowler, 2009; Carter *et al.*, 2009, 2017). These results are consistent with measured pore water pressures beneath Whillans Ice Stream (Blankenship *et al.*, 1987; Engelhardt *et al.*, 1990; Walder and Fowler, 1994; Alley *et al.*, 1997), and radar specularity analysis of present-day Thwaites Glacier, although, in the case of the latter, it is possible that a focused channelised system exists further downstream (Schroeder *et al.*, 2013). These observations may imply that many active subglacial lakes require soft bedded sediments to form and drain (Carter *et al.*, 2017), explaining the tendency for these features to lack the characteristic basal reflection properties used to identify subglacial lakes with radio-echo sounding (Siegert *et al.*, 2015; Carter *et al.*, 2017; Humbert *et al.*, 2018). However, it may be possible to produce, and temporarily sustain, R-channels with cross-sectional areas of ~20 m$^2$ when water is transferred between a system of bedrock overdeepenings by pressure waves that propagate along the length of an ice

stream over several years (Fricker *et al*., 2014; Dow *et al*., 2018), or along ice stream shear margins where basal meltwater

production rates are high (Perol *et al*., 2015; Elsworth and Suckale, 2016; Meyer *et al*., 2016; Bougamont *et al*., 2018).

The extent to which the active subglacial lake drainage events observed in the satellite era are representative of the behaviour of the Antarctic subglacial hydrological system over a full-glacial/interglacial cycle, especially during ice-sheet advance and retreat, is unknown. Differences in discharge magnitude and their association with soft sediments

limit the use of present-day subglacial lake drainage as an analogy for the mechanism which incised channels into hard crystalline bedrock in Pine Island Bay. However, past outburst flooding from subglacial lakes have been suggested to have had a substantial impact on the landscape elsewhere in Antarctica (e.g. Domack *et al*., 2006; Lewis *et al*., 2006; Jordan *et al*., 2010; Larter *et al*., 2019). Jordan *et al*. (2010) proposed that outbursts sourced from a palaeo-subglacial lake with a volume of 850 km$^3$ were responsible for the formation of a series of kilometre-wide bedrock canyons in the Wilkes

Subglacial Basin. This lake is proposed to have accumulated and catastrophically drained during the expansion of the East Antarctic Ice Sheet in the Miocene epoch, at a similar time to when the Labyrinth and the other channel systems in Victoria Land are suggested to have formed (Denton *et al*., 1984, 1993; Sugden *et al*., 1999; Sugden and Denton, 2004; Denton and Sugden, 2005). Morphological evidence for past outburst flood occurrence also exists elsewhere in Scandinavia (Mannerfelt, 1945; Holtedahl, 1967), in North America (Bretz, 1923; Ives, 1958; Wright, 1973) and in

Scotland (Sissons, 1958). The largest of these floods was the release of ~2500 km$^3$ of water during the Missoula floods in eastern Washington State, USA, which have been estimated to have attained peak discharges of ~1.7 x 10$^7$ m$^3$ s$^{-1}$ (Waitt, 1985; O'Connor and Baker, 1992). Although the Missoula floods were released from a proglacial, rather than subglacial, source their estimated volume is similar to the amount of water that could have drained from beneath former Pine Island and Thwaites glaciers if a cascade of upstream lake drainage was to occur (Table 2).


Two models of how Antarctic lakes could form and then drain catastrophically have been hypothesised. Alley *et al*. (2006) proposed a model in which seawater becomes trapped in bedrock basins by the advance and thickening of an ice shelf over the sill of bedrock overdeepenings. Water pressure in the basins would build until the hydraulic potential exceeds that of the next basin downstream, culminating in short-lived, high-magnitude water drainage events, or floods.

An alternative hypothesis was put forward by Jordan *et al*. (2010) in which subglacial water ponds in hydraulic-potential lows produced by the reduced ice-surface slope associated with an advancing ice sheet, with flooding occurring during ice-sheet retreat (Jordan *et al*., 2010). Both of these lake-drainage models occur during either large-scale ice sheet advance or retreat, so are outside of the subglacial hydrodynamics captured by modern observations. Numerical calculations show that these types of event are capable of producing peak discharges of ~10$^5$ m$^3$ s$^{-1}$ (Evatt *et al*., 2006). Our numerical model

results demonstrate that the cascading release of water trapped in a large upstream basin could initiate a flood with a peak discharge of ~5 × 10$^5$ m$^3$ s$^{-1}$. This discharge is approximately equal to the potential carrying capacity of the channels if they were partially ice filled at the time of the flood.

Although high-magnitude fluxes of water could have been released beneath Pine Island and Thwaites glaciers

through cascades of subglacial lake drainage, recent modelling by Beaud *et al*. (2018) suggests that significant erosion can be achieved by relatively low influxes (~20 m$^3$ s$^{-1}$) of seasonal meltwater into the subglacial system when repeated regularly throughout a glacial cycle. For an idealised ice sheet geometry, these simulations demonstrate that small but regular influxes of meltwater are able to excavate channels up to ~20 m deep and 100 m wide into a bedrock substrate over the course of 7500 years (Beaud *et al*., 2018). Although the model omits plucking, which may play a significant role

in the erosion of fractured bedrock by large floods (e.g. Lamb and Fonstad, 2010; Larsen and Lamb, 2016), the substrate

eroded in the model is similar to the hard bedrock present in Pine Island Bay and can therefore be used to assess the formation timescale of the submarine channels. With an average depth of 43 m and a mean width of 507 m, the channels present in Pine Island Bay are substantially larger than the channels that Beaud *et al*. (2018) are able to erode in a single glacial cycle. This supports the interpretation that the huge channels observed beyond the present margins of Pine Island

and Thwaites glaciers must have formed over more than one glacial period. The fluxes of meltwater produced by subglacial lake drainage beneath Pine Island and Thwaites glaciers in the past span discharges similar to those used in the Beaud *et al*. (2018) simulations to several of orders of magnitude more depending on whether the drainage was cascading or only sourced from a single lake. Accordingly, we can envisage a scenario in which recurrent releases of subglacial meltwater trapped in bedrock basins, ranging from frequent gradual drainages to larger outburst floods, could have cut

large channels into bedrock over the course of several glacial cycles. The former lake basins may have acted as sedimentary depocentres which supplied bedload during subglacial lake discharge to enhance the erosive potential of the floods. Thus, when paired with a suitable sediment load for channel erosion, outbursts from a population of subglacial lakes occupying large bedrock basins, occurring either as repeated small outbursts or larger, less frequent, drainage cascades triggered by ice sheet advance or retreat may have been sufficiently erosive to excavate huge channels in the

bedrock of Pine Island Bay.

## 6      Conclusion

The huge palaeo-channel-and-basin systems beyond modern Pine Island and Thwaites glaciers are evidence for

an organised and dynamic subglacial hydrological system beneath former major outlets of the WAIS. Channels of greater dimensions than those in the Labyrinth can only have been incised by large discharges of water flowing under subglacial hydraulic pressure. Our numerical model produces similar steady-state discharges and filling and drainage timescales to those that are observed for subglacial lakes beneath the contemporary Antarctic Ice Sheet. However, the fluxes of water flowing in continuous steady state beneath the LGM ice sheet are too low to have formed channels of the scale observed

in Pine Island Bay. Rather, a higher-magnitude, lower-frequency mechanism is required to explain the formation of the channels. Many mechanisms could have been responsible for channel formation, including propagation of surface melt to the bed and subglacial volcanic eruptions. However, we have argued that on the basis of their geomorphological similarity to features known to have been formed by outburst flooding and their ability to accommodate discharges of at least $2.8 \times 10^5$ m$^3$ s$^{-1}$, even when mostly filled with ice, the most likely candidate for the formation of the Pine Island Bay

channel system was episodic releases of meltwater trapped in upstream subglacial lakes, potentially triggered by ice-sheet advance or retreat. This mechanism may be outside the range of processes captured by modern observations of subglacial hydrology, but is likely to have had an impact on the flow regime of these large ice streams that continue to dominate ice-sheet discharge today. Further observations of the duration and frequency of contemporary subglacial drainage events and the incorporation of more detailed bed topographies into numerical models will help elucidate the role that organised,

episodic outburst flooding plays in the dynamics of the Antarctic Ice Sheet.

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

**Author contributions**

R.D.L., J.A.D. and K.A.H. conceived the study; they and F.O.N. participated in cruises to collect the data. J.D.K. analysed the bathymetry and LIDAR data and conducted the channel measurements. Numerical modelling was conducted by N.S.A., N.R.G., and J.D.K. J.D.K. wrote the initial manuscript with substantial contributions from K.A.H., J.A.D. and R.D.L. All authors contributed to data interpretation and writing of the final manuscript. The authors declare no competing interests.

**Acknowledgements**

We thank the masters and crews of the research vessels, RVIB Nathaniel B. Palmer (NBP), RRS James Clark Ross (JCR) and RVIB Polarstern for their support during data acquisition. Data acquisition using the RRS James Clark Ross was supported by funding from the UK Natural Environment Research Council's iSTAR Programme and grants NE/J005703/1, NE/J005746/1, and NE/J005770/1. Bathymetry data from the expedition ANT-XXIII/4 (https://doi.org/10.1594/PANGAEA.680792) was provided courtesy of the Bathymetry Working Group, Geophysical Department, Alfred Wegener Institute, Helmholtz Centre for Polar and Marine Research. J.D.K was supported by a Debenham Scholarship from the Scott Polar Research Institute, University of Cambridge, and a UK Natural Environment Research Council Ph.D. studentship awarded through the Cambridge Earth System Science Doctoral Training Partnership (grant number: NE/L002507/1). F.O.N. was supported by NSF award 0838735. K.A.H. and R.D.L. were supported by the Natural Environment Research Council – British Antarctic Survey Polar Science for Planet Earth Programme. We thank Povl Abrahamsen for helping to initially clean the iSTAR bathymetry data. We would like to thank Stephen Livingstone and Calvin Shackleton for helpful reviews that improved the paper.

**Data availability**

Marine geophysical data are available on request from the UK Polar Data Centre (https://www.bas.ac.uk/data/uk-pdc/) and the National Centres for Environmental Information (https://www.ngdc.noaa.gov/mgg/bathymetry/multibeam.html) for JCR and NBP data respectively, and on request via PANGAEA (https://pangaea.de/) for Polarstern data. The Dry Valleys LIDAR DEM is available online from the United States Antarctic Resource Center (https://usarc.usgs.gov/lidar_dload.shtml). PISM model outputs are available from N.R.G. upon reasonable request; hydrological model outputs and other datasets analysed during the study are available upon reasonable request from the corresponding author.

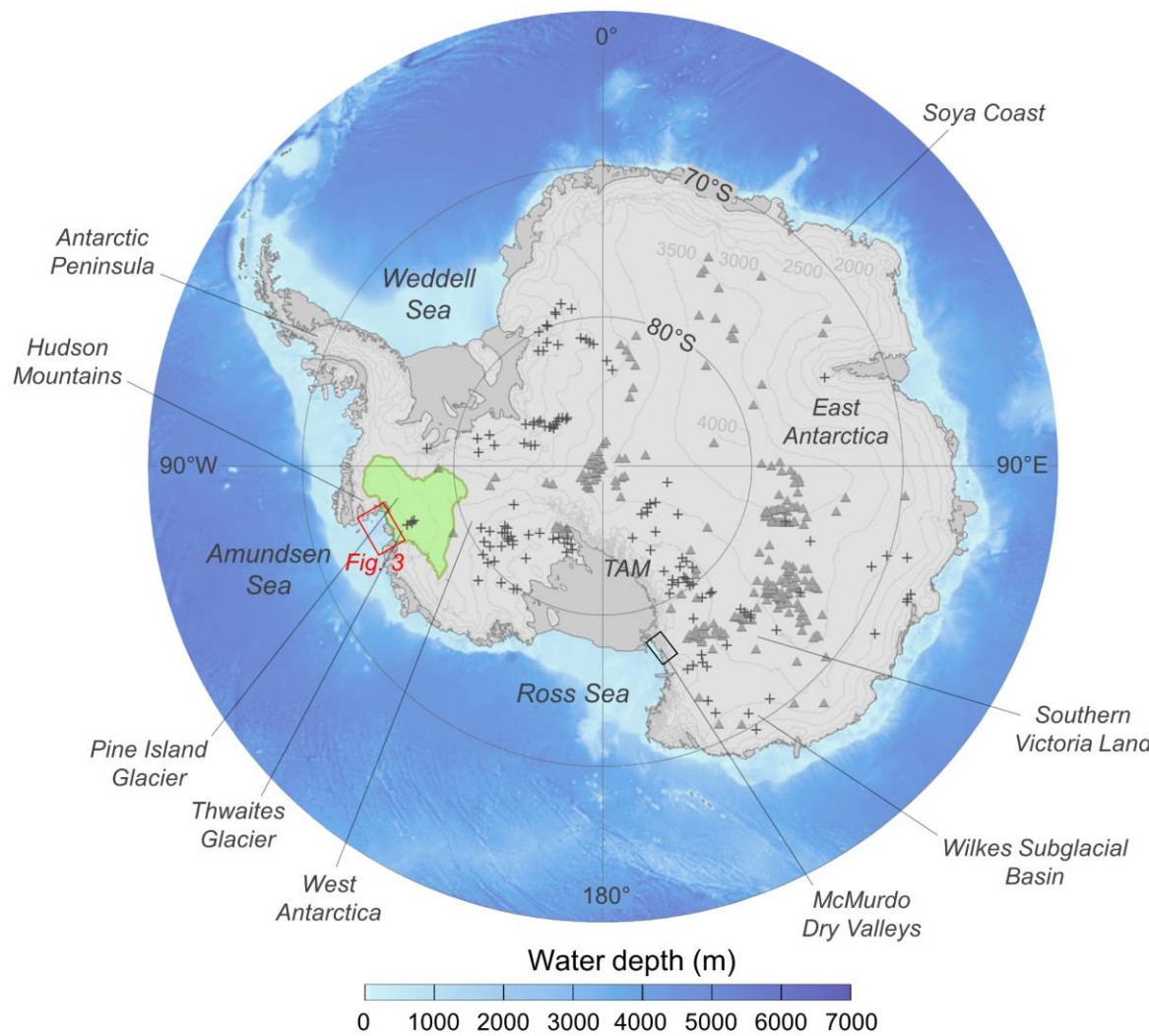

**Figure 1. Overview map displaying the location of features and regions referred to in the text.** The catchment drained by contemporary Pine Island and Thwaites glaciers is highlighted in green. The black box shows the portion of the Transantarctic Mountains (TAM) containing Wright Valley, the Royal Society Range, the Convoy Range and the Asgard Range, in which channel features have been observed. The area shown in Fig. 3 is displayed as a red box. The locations of subglacial lakes observed from radio-echo sounding (grey triangles) and actively draining subglacial lakes observed from satellite altimetry (+) are shown from Wright and Siegert (2012) and Smith *et al.* (2017). Regional bathymetry and ice surface elevation is from BEDMAP 2 (Fretwell *et al.*, 2013).


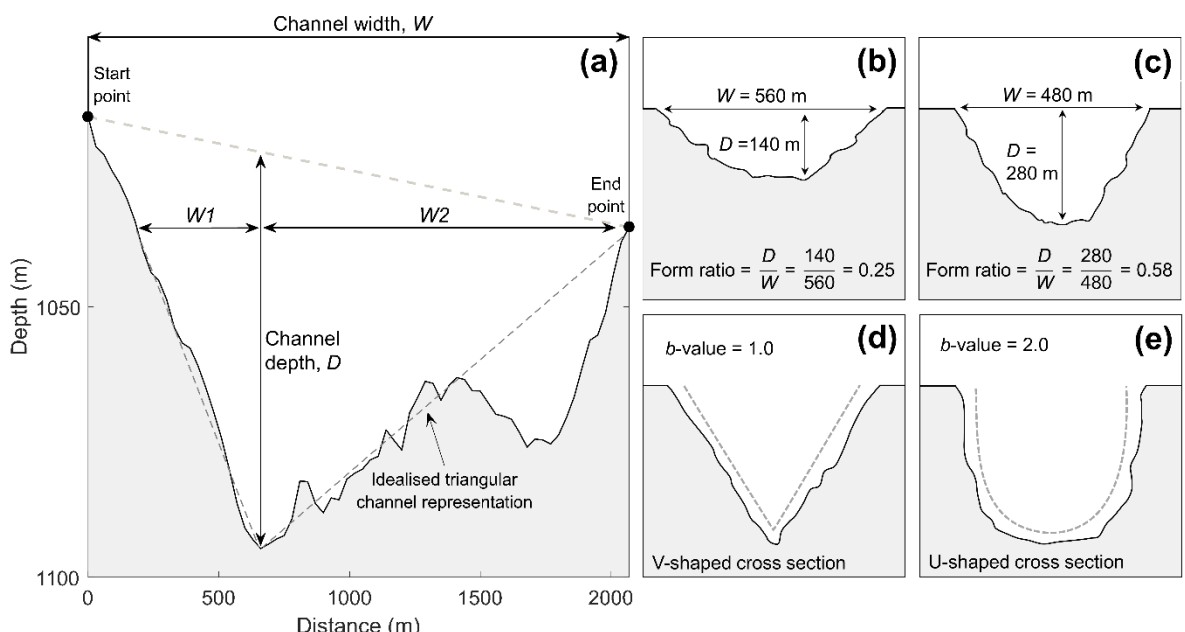

**Figure 2. Methods of quantifying channel morphometry.** (**a**) Output of the semi-automated algorithm used to derive channel cross-sectional metrics, after Noormets *et al*. (2009). Plots (**b**) and (**c**) illustrate the concept of channel form ratio: the ratio of channel depth to channel width. Wider, shallower channels will be characterised by lower form ratios. Plots (**d**) and (**e**) show the variation in channel *b*-value according to channel shape. V-shaped channels will exhibit a *b*-value of ~ 1, whilst U-shaped channels will be characterised by *b*-values of ~ 2. Vertical exaggeration in (**a**) is 20×.



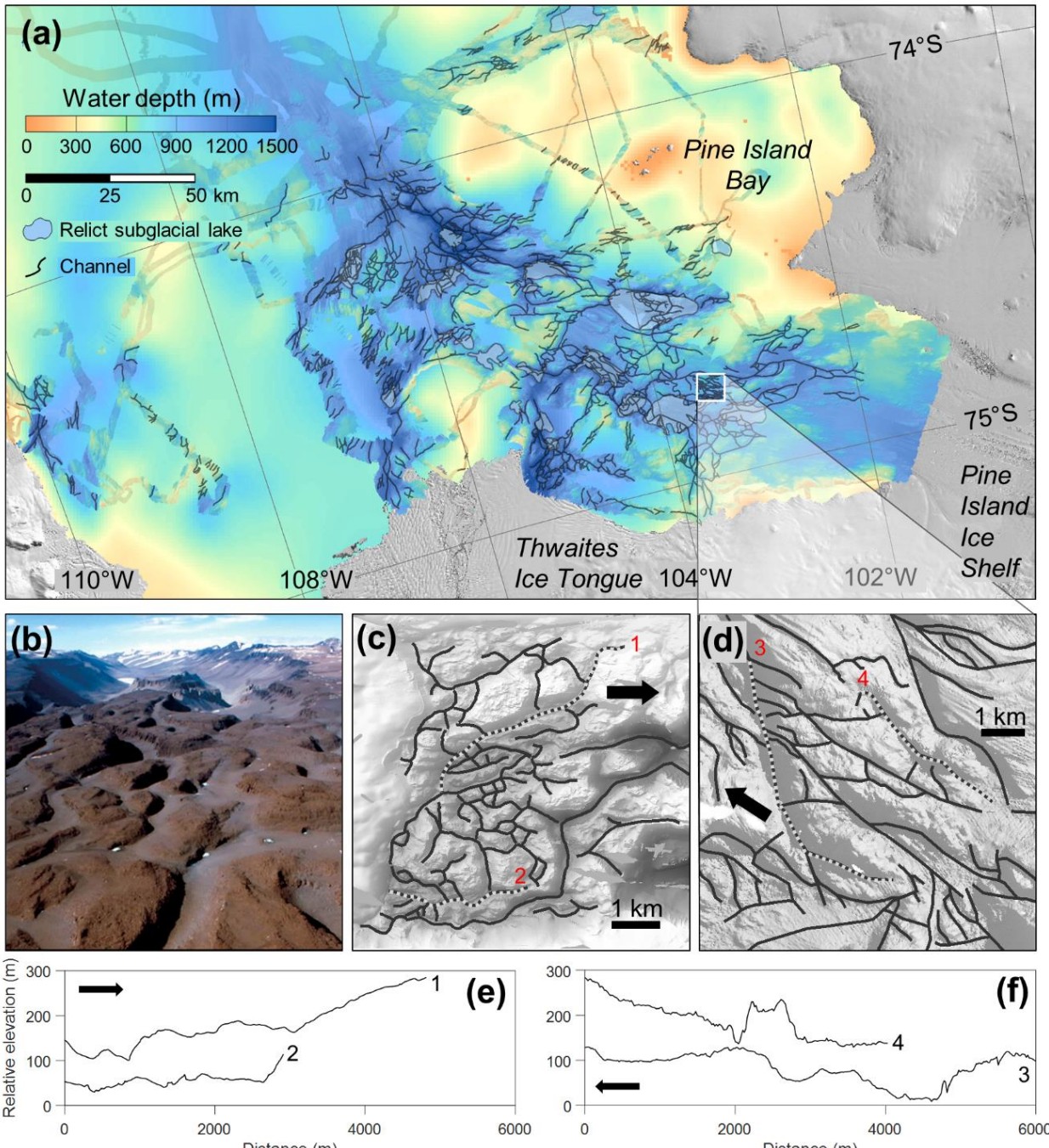

**Figure 3. Channelised bathymetry of the region offshore of Pine Island and Thwaites glaciers.** (**a**) Inner continental shelf bathymetry within Pine Island Bay, gridded at a 20 m cell size with sun illumination from the northeast. Mapped channels are displayed as black lines, and mapped relict subglacial basins are displayed as blue polygons. Onshore topography is displayed as a shaded Landsat Image Mosaic of Antarctica (LIMA) (U.S. Geological Survey, 2007). The white square in (**a**) is scaled to the same dimensions as the Labyrinth, displayed in (**b**) and (**c**). (**b**) Oblique aerial photograph of the channel system comprising the Labyrinth (Lewis *et al*., 2006). Channels in the foreground are ~100 m wide. (**c**) Digital elevation model of the Labyrinth with channels displayed as black lines. (**d**) Subset of the Pine Island Bay bathymetry, scaled to the same dimensions as the Labyrinth, exhibiting a series of channels. Black arrows denote the inferred direction of palaeo-ice flow. Selected long-profiles of the channels comprising the Labyrinth (**e**) and in Pine Island Bay (**f**) with numbers donating their locations in (**c**) and (**d**). Vertical exaggeration in (**e**) and (**f**) is 50 ×.

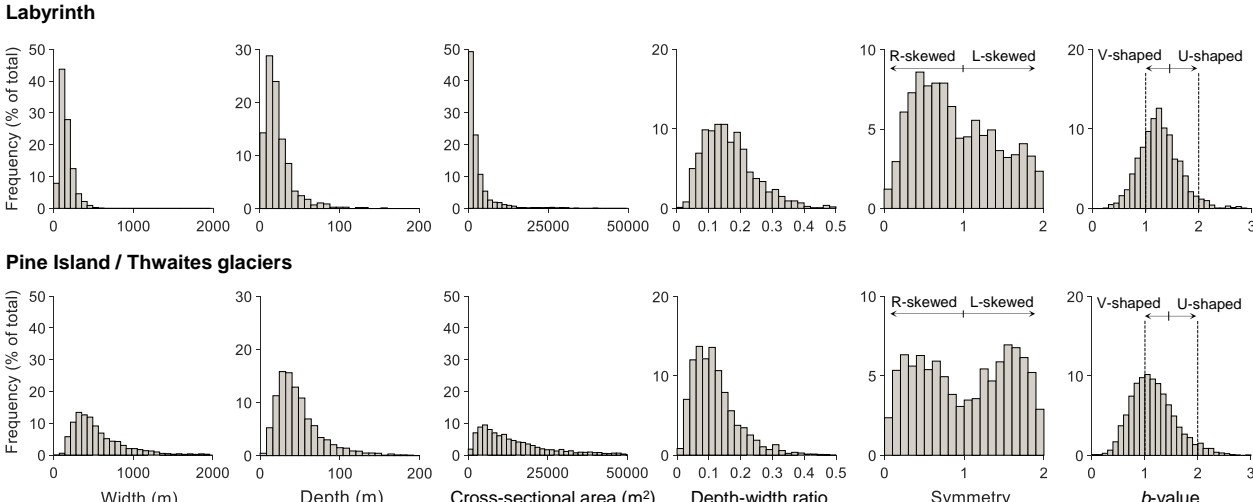

**Figure 4. Size-frequency distributions of the morphometric characteristics of the channels comprising the Labyrinth and those present offshore of Pine Island and Thwaites glaciers.** The channels offshore of Pine Island and Thwaites glaciers from which the distributions in the lower panel are derived are displayed in Fig. 3. The dotted lines in the *b*-value plots correspond to where idealised V-shaped (*b* = 1) and U-shaped (*b* = 2) cross sections would fall on the histogram.

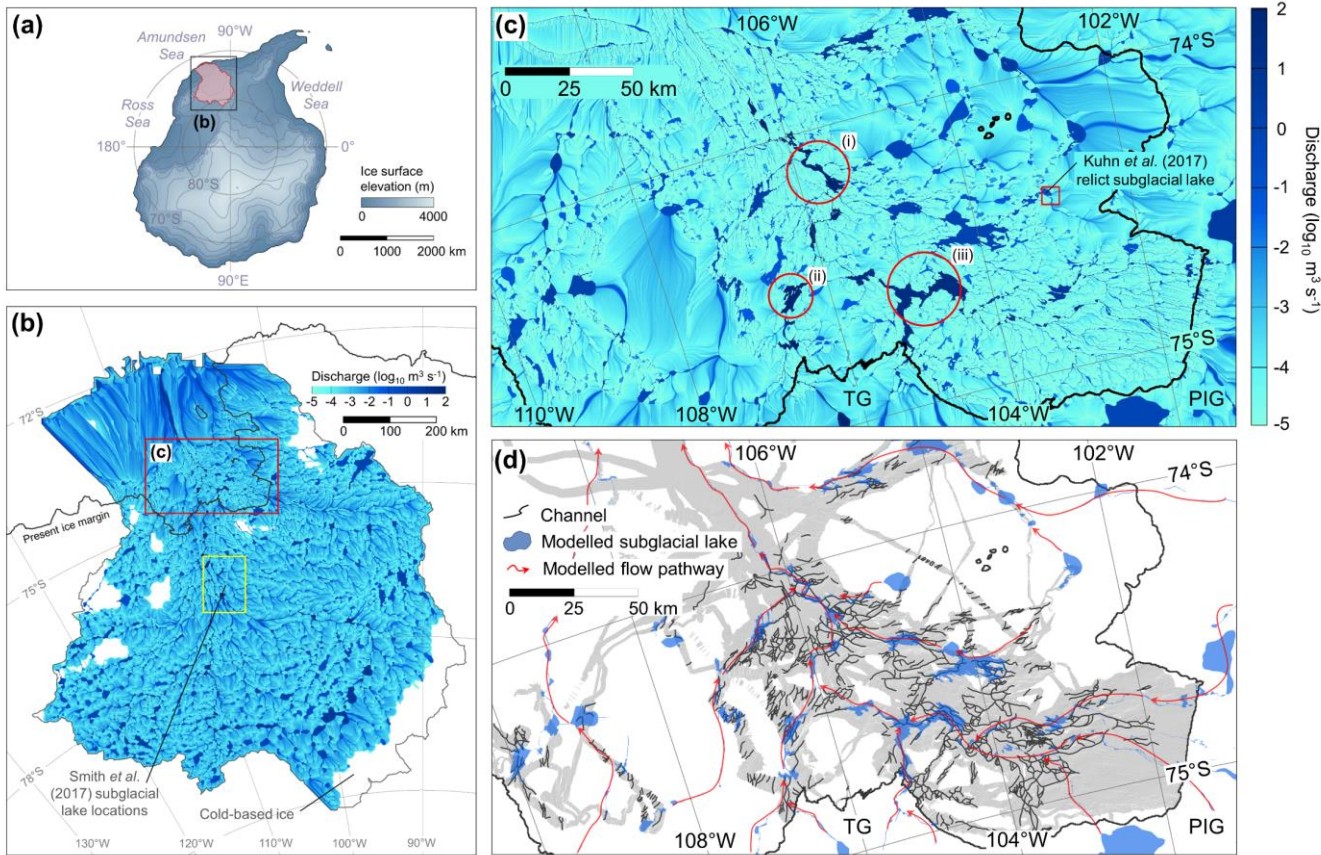

**Figure 5. Modelled water flow beneath Pine Island and Thwaites glaciers at the LGM.** (**a**) Modelled ice sheet surface at 20 ka (Golledge *et al*., 2012), showing the calculated drainage catchment of Pine Island and Thwaites glaciers in red. (**b**) Water discharge and flow routing paths for the entire Pine Island/Thwaites catchment at the LGM, calculated at a 500 m grid resolution. The location of actively filling and draining subglacial lakes beneath the contemporary WAIS (Smith *et al*., 2017) is outlined in yellow. Areas of cold-based ice are displayed in white. (**c**) Modelled subglacial water discharges within the channelised inner shelf region covered by high-resolution (90 m) swath bathymetric data. The locations of four predicted lakes, referred to as examples in the text, are shown. The present-day ice margin from LIMA (U.S. Geological Survey, 2007) is displayed in (**b**), (**c**) and (**d**) as a solid black line. (**d**) Geomorphologically mapped channel network within the high-resolution swath bathymetric data and modelled subglacial drainage network. The major flow routing pathways and subglacial lake locations calculated in (**c**) are displayed as red arrows and blue polygons in (**d**), respectively. Only subglacial lakes with a steady-state discharge >3 m$^3$ s$^{-1}$ are displayed for visual simplicity. The labels 'PIG' and 'TG' mark the contemporary margins of Pine Island Glacier and Thwaites Glacier, respectively.

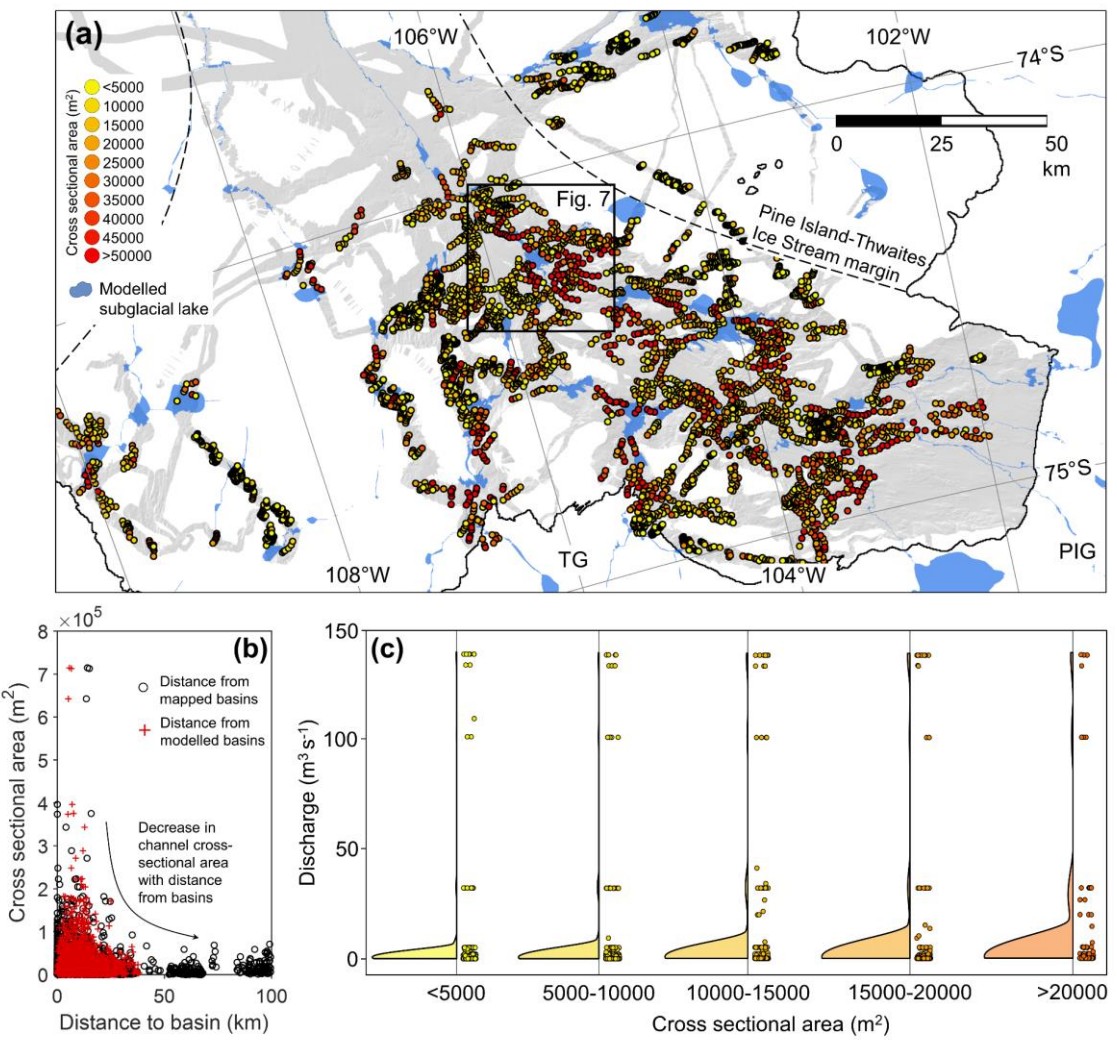

**Figure 6. Spatial variation of channel size within Pine Island Bay.** (**a**) Spatial variation of channel cross-sectional area in relation to the location of modelled subglacial lakes. Channel cross sections used to analyse channel geometry are displayed as dots coloured by channel cross-sectional area. The margins of the former Pine Island-Thwaites Ice Stream (Graham *et al*., 2010) and the location of Fig.7 are shown. The present-day ice margin from LIMA (U.S. Geological Survey, 2007) is displayed as a solid black line. (**b**) Variation in channel cross-sectional area with distance to the nearest mapped and modelled subglacial lake basin. (**c**) Raincloud plot showing the data and distributional shape of modelled steady-state discharges through channels of different cross-sectional areas. Rainclouds were produced using code from Allen *et al*. (2019).

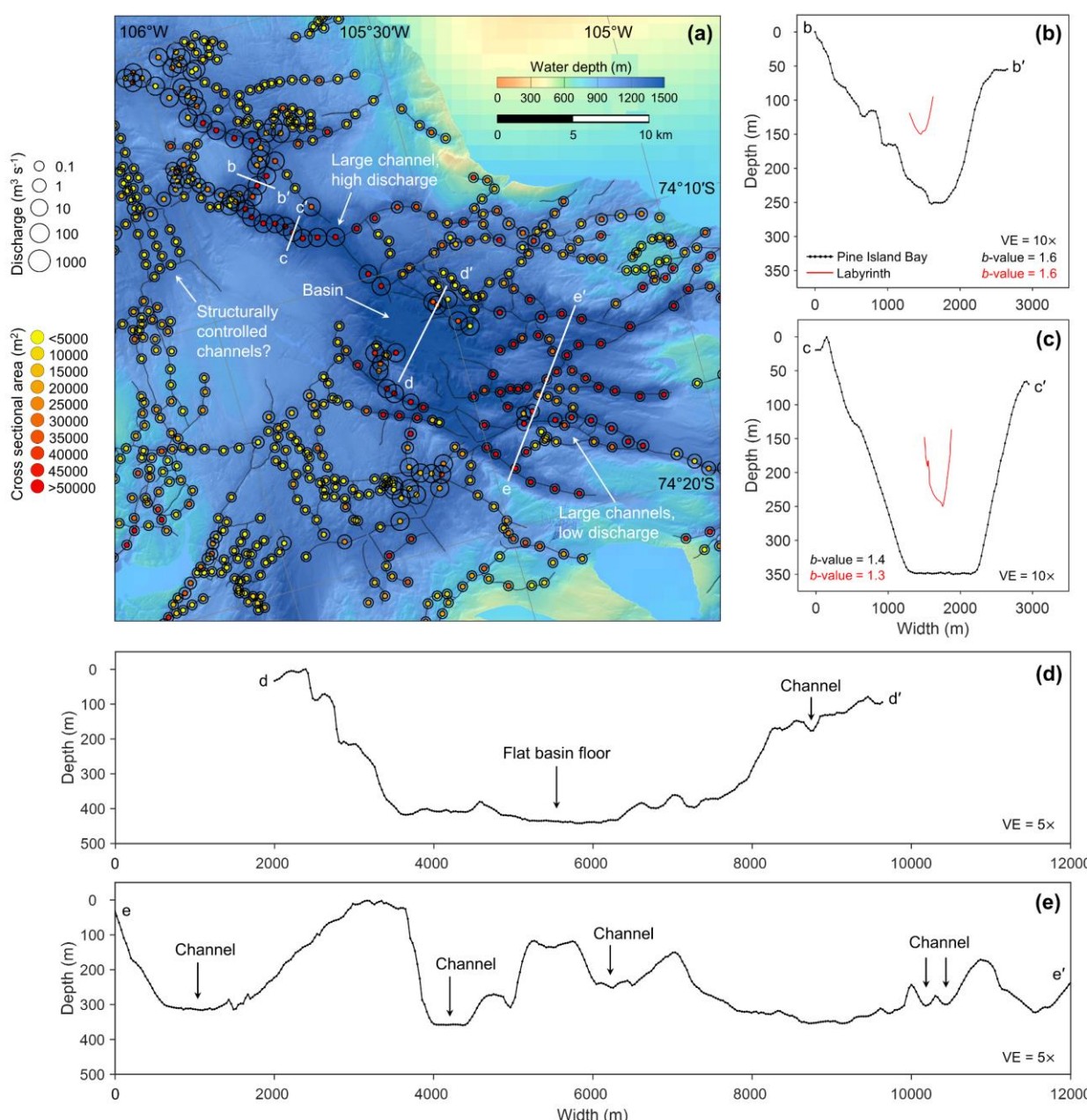

**Figure 7. Relationship between modelled water flux and channel size through a channel-basin system.**

(**a**) Example channel-basin system in Pine Island Bay. Mapped channels are displayed as black lines. Cross sections used to calculate channel geometries are displayed as dots coloured by channel cross-sectional area. The size of the circle surrounding the dots represents the calculated steady-state water flux through the channels. The locations of the cross profiles displayed in **b-e** are shown. Note the contrast between the large, high discharge channel to the northwest of the basin and the equally large, but low discharge, channels flowing into the basin from the southeast. (**b-c**) Cross-sectional profiles of a large channel in Pine Island Bay compared to some of the largest channels in the Labyrinth. (**d**) Cross section of a basin in Pine Island Bay. (**e**) Cross section of a series of channels feeding the basin from the southeast. Vertical exaggeration (VE) is 10× in (**b-c**) and 5× in (**d-e**).

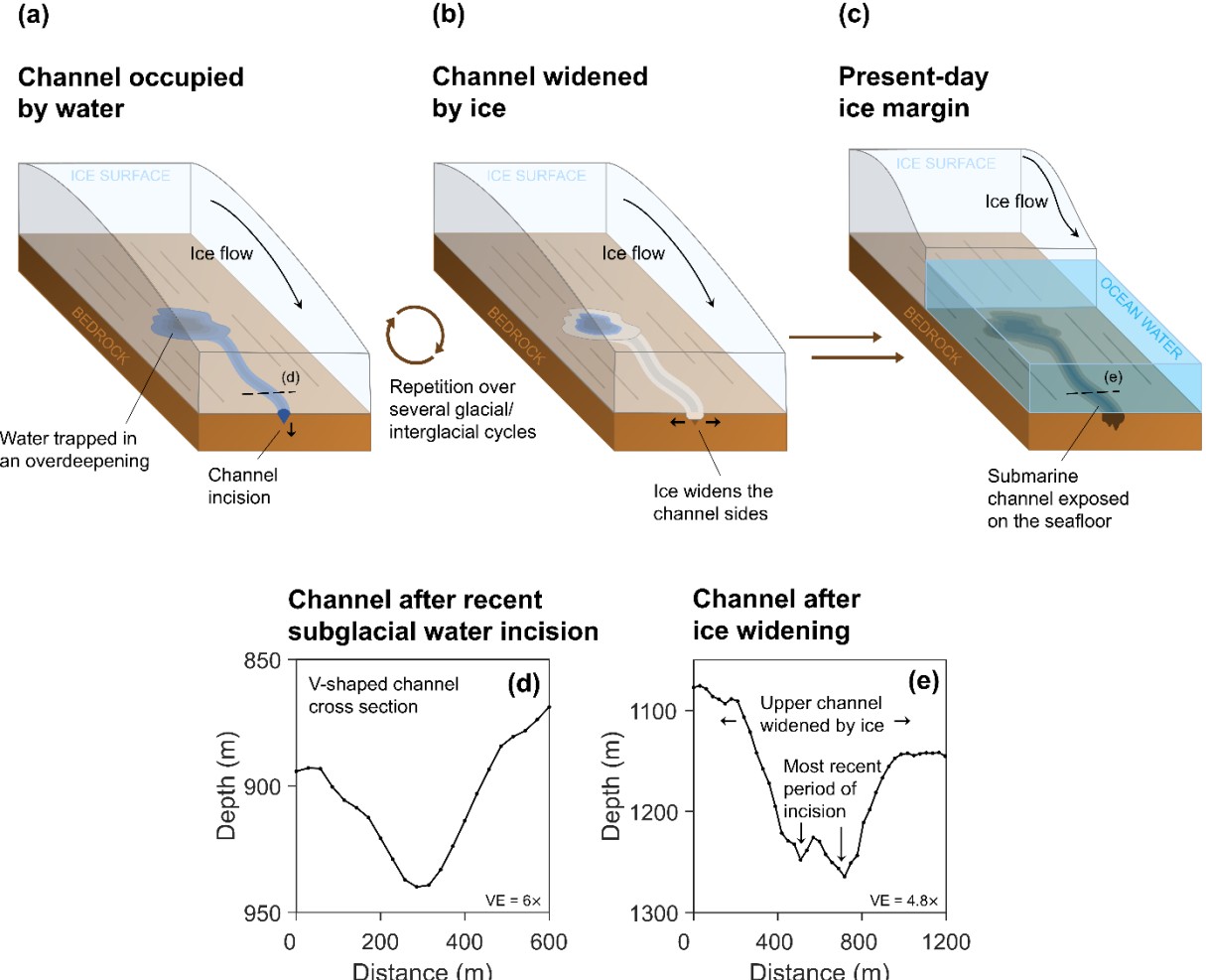

**Figure 8. Schematic of channel formation over multiple glacial-interglacial cycles. (a)** Channels are incised by short-lived episodes of fast-flowing subglacial meltwater, potentially released from lakes trapped in overdeepenings during ice sheet advance and retreat. **(b)** Repeated periods of ice overriding widens the top of the channels. **(c)** A composite channel feature, produced over multiple glaciations, is left exposed on the seafloor. **(d)** Example of a more recently incised channel, characterised by a V-shaped cross section. **(e)** Cross section of a channel persisting from previous glacial periods containing both V-shaped meltwater signatures and a wide upper channel section eroded by ice. Vertical exaggeration (VE) is 6× in **(d)** and 4.8× in **(e)**.

**Table 1.** Details of the geophysical cruise data used in this investigation.

| Cruise / ID | Ship | Year | Multibeam system | Principal investigator |
|---|---|---|---|---|
| ANT-XXIII/4 | Polarstern | 2006 | Hydrosweep DS2 | Gohl, K. |
| JR179 | James Clark Ross | 2008 | EM120 | Larter, R. |
| JR294 | James Clark Ross | 2014 | EM122 | Heywood, K. |
| NBP0001 | N.B. Palmer | 2000 | SeaBeam 2112 | Jacobs, S. |
| NBP0702 | N.B. Palmer | 2007 | EM120 | Jacobs, S. |
| NBP0901 | N.B. Palmer | 2009 | EM120 | Jacobs, S. |
| NBP1210 | N.B. Palmer | 2013 | EM120 | Halanych, K. |
| NBP9902 | N.B. Palmer | 1999 | SeaBeam 2112 | Anderson, J. |

**Table 2.** Volume, filling time, local steady-state discharge, and potential mean flood discharge of lakes displayed in Fig. 5.

| Lake | Local steady-state water discharge ($m^3\,s^{-1}$) | Lake volume ($km^3$) | Time to fill (years) | Local potential mean discharge ($m^3\,s^{-1}$)[*] | Total upstream volume ($km^3$) | Potential mean flood discharge[**] ($m^3\,s^{-1}$) |
|---|---|---|---|---|---|---|
| (i) | 139 | 3.29 | 0.73 | 78.2 | 2762 | 65690 |
| (ii) | 101 | 3.47 | 1.09 | 82.5 | 2329 | 55390 |
| (iii) | 32.2 | 19.9 | 19.6 | 473 | 389 | 9251 |
| (iv)[***] | 0.23 | 0.33 | 45.1 | 7.8 | 0.58 | 13.8 |

[*]Local mean discharge calculated by assuming drainage of the lake in 16 months, assuming no further inflow into the basin during drainage (c.f. Wingham *et al*. 2006).
[**]Potential mean flood discharge calculated by assuming drainage of a large upstream lake in 16 months (c.f. Wingham *et al*. 2006) which triggers the cascading drainage of all lakes downstream from that initial event.
[***]Basin suggested to be a relict subglacial lake by Kuhn *et al*. (2017).