# Peer review of "Past water flow beneath Pine Island and Thwaites glaciers, West Antarctica"

_The Cryosphere, 2019_

## Referee Comment (RC1) · Stephen Livingstone (Referee) · 6 May 2019

**Kirkham et al. (2019) Past water flow beneath Pine Island and Thwaites glaciers, West Antarctica. The Cryosphere Discussions**

**Stephen Livingstone**

**General Comments**

This paper presents new mapping of over 1000 subglacial channels and basins exposed by the retreat of Pine Island and Thwaites glaciers since the LGM. The distribution and morphology of these channels are analyses and compared with the Labyrinth channels. To assess the volumes and routing of subglacial water beneath the Pine Island and Thwaites system, hydrological modelling of LGM conditions is utilised. The methods appear to be generally robust and appropriate and the results are detailed and address the important challenge of identifying how large (kms scale) bedrock cut channels/tunnel valleys form. Using these data, the authors favour an origin by episodic high magnitude discharges from subglacial lakes. The paper is well written and the figures generally clear and informative. I certainly favour publication of this paper. However, I do have a number of points that should be considered, in particular the four below under general comments.

1. Relationship between channel metrics and modelled discharges. Although the morphometric analysis presented in Figure 4 is really informative, it would have been nice to also see the spatial variation in channel size analysed and presented. Given the authors have already calculated variations in the potential mean discharges and flow discharges for four potential subglacial lakes on the palaeo-ice stream bed using hydrological modelling, it would be relatively straightforward to also compare how the local morphologies of the channels in these different regions relate to these findings. If the channels were cut by cascading lake drainages, you might expect a scale association between flood discharges (from your model) and the size/morphology of your mapped channels. Put simple, are the larger channels associated with regions associated with the largest discharges? As well as looking at your particular areas, you could compare the average size of channels in the many tributaries, to the channels in the main channel, where they converge. If an association can be shown, it would be really strong evidence that your interpretation is correct, and I recommend you carry out this additional analysis.

2. Exploration of less catastrophic drainages – In lines 486-510 you argue against multiple small subglacial lake drainage events. Given your recharge times are on the order of years to decades, there is the potential for high frequency (1000s of events), small-medium magnitude subglacial lake drainage events over multiple glacial cycles. The authors seem to discard this possibility too easily, especially given some recent work by Beaud et al. (2018) who suggest that bedrock channels on the scale of tunnel valleys can be excavated over several thousand years from seasonal meltwater discharge. Could relatively small yearly to decadal drainage of subglacial lakes produce a similar distribution of channels over a glacial cycle or two, especially given the large catchment area and significant volume of water modelled to be contained in lakes upstream of the channels?

*- Beaud, F., Venditti, J.G., Flowers, G.E. and Koppes, M., 2018. Excavation of subglacial bedrock channels by seasonal meltwater flow. Earth Surface Processes and Landforms, 43(9), pp.1960-1972.*

3. Channel cross-section - Looking at the picture of the Labyrinth channels in Fig. 3B, I am surprised that they do not come out as more U-shaped. It would be nice to see a few different examples of 'typical' cross-profiles for the two regions so the reader can relate the b-index numbers to a

tangible cross-profile. I also wonder what b-index values a trapezoid channel would produce? In general, I have typically linked mega-flood events (in subaerial settings at least) to more canyon-like channel forms (e.g. Channelised Scablands – Bretz references in paper; mega-flood channels in the English Channel – e.g. Gupta et al. 2007). The tendency towards V-shaped forms in both settings should be discussed in relation to the above literature given your interpretation.

*- Gupta, S., Collier, J.S., Palmer-Felgate, A. and Potter, G., 2007. Catastrophic flooding origin of shelf valley systems in the English Channel. Nature, 448(7151), p.342.*

4. Bedrock Geology – This might not be possible, but do you have any clue about the geology of the bed that could be used to help frame your discussion – e.g. is the bedrock hard or soft?

**Specific Comments**

L49: Also see Siegfried, M.R., Fricker, H.A., Carter, S.P. and Tulaczyk, S., 2016. Episodic ice velocity fluctuations triggered by a subglacial flood in West Antarctica. Geophysical Research Letters, 43(6), pp.2640-2648.

L74: It is odd to reference Denton and Sugden and Evatt here given you are referring to Glacial Lake Missoula. Also, although I think it is fine to reference this flood as an example of the scale of mega-floods, that this flood was proglacial not subglacial does not make the comparison entirely fair. Do we expect pressured subglacial floods to behave in the same manner as proglacial outbursts? This difference should be made clearer.

L83: I agree here that it is inconsistent with a single/ few drainage events. But if given enough time could these not eventually produce significant channels?

L135: In the example in Figure 2a the channel illustrated is a composite feature (and also shown in Figure 6e). How would this effect the b-index? I think some discussion of this would be appropriate in the methods; it would also be useful to describe how common this configuration is in the results.

L253: Refer to Figure 4.

L255: The 19 flat-bottomed depressions are not mapped in Figure 3.

L257: The channels also appear to run through the basins.

L265: Need to refer to Figure 4.

L274-277: I like this, but am then surprised that this difference is not apparent in the form ratio (depth: width). Is there an explanation for this?

L345-346: Or could it be a composite meltwater signature formed over a long time. You seem to pre-empt your discussion here, and I suggest initially presenting every possible scenario.

L366-369: An alternative explanation for the Figure 6e cross-profile is it is a U-shaped channel cut by ice that has then had meltwater erosion at the base cutting the two smaller v-shaped channels. Can this scenario be ruled out?

L377: I don't follow the logic of this sentence with respect to the last – if there is enough water and a crack to initiate hydrofracture water will get to the bed.

L415: "with observed discharges of..."

L428: I would not include the Bretz references here as you are referring to subglacial channel formation. I also suggest you remove the British-Irish Ice Sheet references as there is a scale difference. Some of the classic literature on bedrock cut tunnel valleys could also be referred to here (e.g. Boyd et al., 1988; Mullins and Hinchey, 1989; Kor et al., 1991; Brennand and Shaw).

*- Brennand, T.A. and Shaw, J., 1994. Tunnel channels and associated landforms, south-central Ontario: their implications for ice-sheet hydrology. Canadian Journal of Earth Sciences, 31(3), pp.505-522.*

*- Kor, P.S.G., Shaw, J. and Sharpe, D.R., 1991. Erosion of bedrock by subglacial meltwater, Georgian Bay, Ontario: a regional view. Canadian Journal of Earth Sciences, 28(4), pp.623-642.*

*- Mullins, H.T. and Hinchey, E.J., 1989. Erosion and infill of New York Finger Lakes: Implications for Laurentide ice sheet deglaciation. Geology, 17(7), pp.622-625.*

L437: Can you quantify the erosion rate?

L464: Could also cite: *Livingstone, S.J., Chu, W., Ely, J.C. and Kingslake, J., 2017. Paleofluvial and subglacial channel networks beneath Humboldt Glacier, Greenland. Geology, 45(6), pp.551-554.*

L466: See also the above reference and *Cooper, M.A., Michaelides, K., Siegert, M.J. and Bamber, J.L., 2016. Paleofluvial landscape inheritance for Jakobshavn Isbræ catchment, Greenland. Geophysical Research Letters, 43(12), pp.6350-6357.*

L469: See the Livingstone et al. (2017) reference for an example of subglacial channels beneath the Greenland Ice Sheet. I also wonder whether this result is a function of the different setting of Greenland – for instance, infilling of valleys/fjords with sediments, potentially burying valleys; and less (and less detailed) sea-floor mapping.

L486-503: How do you reconcile this paragraph with that on lines 512-523, where you cite multiple papers presenting palaeo-evidence for active subglacial lakes and drainages across bedrock?

L522: "drained" might be a better word than "utilised" here.

**Figures**

Figure 3 – You also state that you identify 19 former lake basins. It would be useful to include these here if mapped? Or are these based on the modelling results?

Figure 4 – It is not clear what the dotted lines and the arrows in the V/U shaped plots refer too.

Figure 5 – Plot (d) rather reproduces Figure 3a. Could you have included the basins in Fig. 3 and rather overlay the modelled basins and channels here so the reader can directly compare how they match up. For (b) and (c) I would find it helpful if the current grounding line was included to help give some context to the reader.

---

## Referee Comment (RC2) · Calvin Shackleton (Referee) · 15 May 2019

**Past water flow beneath Pine Island and Thwaites glaciers, West Antarctica**

Kirkham et al. 2019

Review: Calvin Shackleton

**General comments:**

This manuscript presents new mapping and morphometric analysis of subglacial channels and basins on the seafloor exposed by the retreating Pine Island and Thwaites glaciers, West Antarctica. The work also utilises a modelled LGM ice surface and isostatically corrected bed topography to model past water flow, with particular focus on water production and storage. The methodology for morphometric analysis of channels is thorough and well-explained, and the modelling approach is justified appropriately. It should perhaps be noted in the methods that the model approach does not allow for the prediction of anastomosing channels, and should be/is used only to predict water flow direction rather than simulate the behaviour of individual channels.

The results section is concise, and provides select relevant metrics from what was undoubtedly a large dataset. The figures are useful and informative, although a detailed map of the interpreted subglacial basins is currently lacking and could be incorporated into Figure 3, along with some long- and cross-profiles of the basins. This manuscript is very well written and addresses an important topic in subglacial hydrology, with an interesting discussion of the origin and cyclic behaviour of high-magnitude subglacial lake drainage events and their impacts on subglacial hydrology and landscape development. I think this work should be published and I propose a small number of minor suggestions and corrections to improve the manuscript.

**Specific comments:**

L247: I like this interpretation, but it would be nice to see some evidence of the "lines of geological weakness" for comparison to the channels.

L255-259: Refer to the appropriate figure that you based your descriptions and interpretations on. At the moment I can't find a figure where the basins are clearly mapped, and suggest that a detailed description of the basins should be included, supported by select long- and cross-profiles that could be incorporated into figure 3.

L344-347: This could be true, or they could have been formed over a longer period, potentially over multiple glacial cycles. Is there any other evidence that can be presented here that leads you to favour formation by large volumes of subglacial water? Or to rule out formation over longer periods?

L347-354: What if the channels were widened (and/or deepened) by sliding ice following infilling/ channel closure during their inactive phase (i.e. seasonally) rather than during subsequent ice advance/retreat?

L375: This interpretation needs to be justified a little better. Why would surface water **not** reach the bed? Are there no crevasses in these regions? Can you rule out hydrofracture? To me the

documentation of surface meltwater rivers by Bell and Kingslake conversely indicates a high potential for surface meltwater entering the englacial and basal system.

L494: Insert comma after "Thwaites Glacier"

L519: Given that this paragraph is discussing floods from subglacial lakes, the comparison to proglacial lake Missoula seems a little out-of-place. It should be made clear that this is a proglacial lake and the comparison you are making is between their high-discharge rather than drainage environment.

L879: "Bindschadler" missing an r

L1227: Indicate that the long profiles are from **several/select** channels in the Labyrinth and Pine Island Bay region.

**Figures:**
*Figure 3:*

I cannot see an inset map as described in the caption. Are you referring to the labelling in figure 1?

It is a little confusing to have the labels on the downstream end in 3c and upstream end in 3d, I suggest to have labels only at downstream end to match the profiles in 3e and 3f below.

*Figure 5:*

Perhaps include a grid on 5a to help orient readers who are not used to looking at projections of Antarctica.

Here it would be nice to show the overlap between modelled basins and geomorphologically mapped basins (which I think should also be presented in figure 3).

---

## Author Comment (AC1) · 17 Jun 2019

Final author comments on reviews of Kirkham et al. (2019): Past water flow beneath Pine Island and Thwaites glaciers, West Antarctica. The Cryosphere Discussions.

We thank Stephen Livingstone and Calvin Shackleton for their constructive reviews. Their comments have helped to improve the paper. Below, we set out how we have responded to the reviewer comments. Each of the original reviewer comments is first repeated, followed by an explanation of how we have dealt with the comment. We also attach a track changes version of the revised manuscript as a supplement to this comment.

Reviewer 1: Stephen Livingstone

[Figure]

General Comments

This paper presents new mapping of over 1000 subglacial channels and basins exposed by the retreat of Pine Island and Thwaites glaciers since the LGM. The distribution and morphology of these channels are analysed and compared with the Labyrinth channels. To assess the volumes and routing of subglacial water beneath the Pine Island and Thwaites system, hydrological modelling of LGM conditions is utilised. The methods appear to be generally robust and appropriate and the results are detailed and address the important challenge of identifying how large (kms scale) bedrock cut channels/tunnel valleys form. Using these data, the authors favour an origin by episodic high magnitude discharges from subglacial lakes. The paper is well written and the figures generally clear and informative. I certainly favour publication of this paper. However, I do have a number of points that should be considered, in particular the four below under general comments.

1. Relationship between channel metrics and modelled discharges. Although the morphometric analysis presented in Figure 4 is really informative, it would have been nice to also see the spatial variation in channel size analysed and presented. Given the authors have already calculated variations in the potential mean discharges and flow discharges for four potential subglacial lakes on the palaeo-ice stream bed using hydrological modelling, it would be relatively straightforward to also compare how the local morphologies of the channels in these different regions relate to these findings. If the channels were cut by cascading lake drainages, you might expect a scale association between flood discharges (from your model) and the size/morphology of your mapped channels. Put simple, are the larger channels associated with regions associated with the largest discharges? As well as looking at your particular areas, you could compare the average size of channels in the many tributaries, to the channels in the main channel, where they converge. If an association can be shown, it would be really strong evidence that your interpretation is correct, and I recommend you carry out this additional analysis.

We conducted the analysis suggested by analysing a further 2000 cross-sections (making a total of >4200 channel samples in Pine Island Bay). We have created an additional two figures (Figures 6 and 7) which examine the relationships between channel metrics and modelled discharges through the channels. In the revised manuscript, lines 301-313 discuss the relationship between channel size (we chose to present cross sectional area as it is a function of both width and depth) and modelled steady state water discharges. We found that larger channels are generally associated with higher steady state discharges and occur in close proximity to the basins (Fig. 6), whilst smaller channels are generally located further out from the mapped and modelled basins and are associated with lower steady state discharges. We found no relationship between modelled discharge and channel shape (symmetry or b-value).

However, although larger channels are associated with higher discharges, this does not hold true in all cases as some of the smallest channels (cross sectional area <5000 m2) carry some of the highest steady state discharges ($\sim$130 m3 s-1) whilst many of the largest channels are predicted to contain discharges <0.1 m3 s 1 (Figure 6c). To demonstrate this, we present an example in Figure 7 where four channels that are an equivalent size contain discharges over three orders of magnitude less than the main outflow channel from the basin into which the four channels feed. This result supports the interpretation that the channels were not formed by the continuous flow of water beneath former Pine Island and Thwaites glaciers as, if this was the case, we would expect all large channels to be associated with a high discharge. Similarly, in some instances, some of the smallest channels have some of the highest discharges. The implications of this for channel genesis are discussed in lines 432-436 and lines 522-524, which argue that the fact that the largest channels are located close to basins (Figure 6b) strongly supports the notion that channel genesis is connected to the basins, likely through episodic outbursts from the subglacial lakes which formed the channels.

2. Exploration of less catastrophic drainages – In lines 486-510 you argue against multiple small subglacial lake drainage events. Given your recharge times are on the

order of years to decades, there is the potential for high frequency (1000s of events), small-medium magnitude subglacial lake drainage events over multiple glacial cycles. The authors seem to discard this possibility too easily, especially given some recent work by Beaud et al. (2018) who suggest that bedrock channels on the scale of tunnel valleys can be excavated over several thousand years from seasonal meltwater discharge. Could relatively small yearly to decadal drainage of subglacial lakes produce a similar distribution of channels over a glacial cycle or two, especially given the large catchment area and significant volume of water modelled to be contained in lakes upstream of the channels?

(Beaud, F., Venditti, J.G., Flowers, G.E. and Koppes, M., 2018. Excavation of subglacial bedrock channels by seasonal meltwater flow. Earth Surface Processes and Landforms, 43(9), pp.1960-1972.)

We thank the reviewer for bringing this paper to our attention. It is important to note that although some of the steady state discharges calculated by our model are the same order of magnitude, and in some instances higher, than the water fluxes used by Beaud et al. (2018), the vast majority are much lower and would be incapable of transporting coarse bedload. Combined with the discrepancy between channel cross sectional area and modelled discharge (see our reply to general comment 1), it is clear that steady state discharges did not form the channels, even if subglacial water is more erosive than previously supposed.

However, we agree that in light of these results, lower magnitude discharges from subglacial lakes may have had a significant erosive role in excavating the channels. Accordingly, lines 579-596 now outline the recent Beaud et al. (2018) work and discuss its implications for the formation of the channels in Pine Island Bay. The main conclusions outlined in this discussion are:

1) That the Beaud et al. (2018) modelling study supports our interpretation that the channels in Pine Island Bay formed over multiple glacial cycles.

2) Our modelled lake drainage fluxes range from similar to those used in the Beaud et al. simulations to several orders of magnitude greater, depending on whether the lake drainage occurred alone or was triggered as a drainage cascade.

3) The lake basins may have acted as sedimentary depocentres that provided bedload during lake drainage, facilitating greater erosion.

We therefore conclude that: "when paired with a suitable sediment load for channel erosion, outbursts from a population of subglacial lakes occupying large bedrock basins, occurring either as repeated small outbursts or larger, less frequent, drainage cascades triggered by ice sheet advance or retreat may have been sufficiently erosive to excavate huge channels in the bedrock of Pine Island Bay."

3. Channel cross-section - Looking at the picture of the Labyrinth channels in Fig. 3B, I am surprised that they do not come out as more U-shaped. It would be nice to see a few different examples of 'typical' cross-profiles for the two regions so the reader can relate the b-index numbers to a tangible cross-profile. I also wonder what b-index values a trapezoid channel would produce? In general, I have typically linked mega-flood events (in subaerial settings at least) to more canyon like channel forms (e.g. Channelised Scablands – Bretz references in paper; mega-flood channels in the English Channel – e.g. Gupta et al. 2007). The tendency towards V shaped forms in both settings should be discussed in relation to the above literature given your interpretation.

(Gupta, S., Collier, J.S., Palmer-Felgate, A. and Potter, G., 2007. Catastrophic flooding origin of shelf valley systems in the English Channel. Nature, 448(7151), p.342.)

We have produced an additional figure (7) for the manuscript which compares a select number of cross sections from Pine Island Bay to those of the Labyrinth. We chose to compare some of the largest channels from each of the two areas to emphasise the difference in scale between the Labyrinth and the channels in Pine Island Bay. The four cross sections in Figure 7b and Figure 7c have their associated b-values stated in the figure. The examples demonstrate that the Labyrinth and Pine Island Bay channels

have very similar cross sectional shapes, although the Labyrinth channels are slightly more U shaped overall (Figure 4). As displayed in Figure 7c, trapezoidal cross sections tend to reflect a combination of V and U shaped geometries and typically have b-values of ∼1.3-1.6. However, it needs to be noted that in Pine Island Bay, the bases of some of the channels are infilled with sediment, which tends to give a more box shaped profile that may not reflect the true morphology of the channel. This is stated in line 358. We now discuss how the shape of the channels affects our interpretation of their formative process in lines 351-359 which read: "Channel form ratios and b-values demonstrate that both sets of channels tend to have broad and shallow V-shaped cross sectional profiles (Figure 4), indicative of subglacial meltwater erosion, rather than the U shaped morphology associated with direct glacial erosion (Pattyn and Van Huele, 1998; Rose et al., 2014). Some of the channels also have trapezoidal cross sectional forms (b-values ∼1.3-1.6; Figure 7c). This cross sectional shape has been associated with high magnitude discharges of water (e.g. Bretz, 1923, 1969; Gupta et al., 2007; Larsen and Lamb, 2016), although the shape of the channels in Pine Island Bay, and accordingly their b values, may be unrepresentative where significant sediment infill of the base of the channel is present (e.g. Smith et al., 2009)."

4. Bedrock Geology – This might not be possible, but do you have any clue about the geology of the bed that could be used to help frame your discussion – e.g. is the bedrock hard or soft?

We have added lines 368-370 to remedy this point. The lines state: "The inner shelf substrate in Pine Island Bay consists mainly of hard granitoid bedrock and porphyritic dykes (Pankhurst et al., 1993; Kipf et al., 2012; Gohl et al., 2012; Lindow et al., 2016) that would be resistant to erosion by subglacial meltwater." Lines 579-585 also compare the strength of the Pine Island Bay bedrock to that used in the Beaud et al. (2018) simulations to help consider possible timescales of channel erosion.

Specific Comments

L49: Also see Siegfried, M.R., Fricker, H.A., Carter, S.P. and Tulaczyk, S., 2016. Episodic ice velocity fluctuations triggered by a subglacial flood in West Antarctica. Geophysical Research Letters, 43(6), pp.2640-2648.

Reference added.

L74: It is odd to reference Denton and Sugden and Evatt here given you are referring to Glacial Lake Missoula. Also, although I think it is fine to reference this flood as an example of the scale of megafloods, that this flood was proglacial not subglacial does not make the comparison entirely fair. Do we expect pressured subglacial floods to behave in the same manner as proglacial outbursts? This difference should be made clearer.

We have removed the Evatt reference here and have moved the Denton and Sugden reference to earlier in the paragraph where the channels in southern Victoria Land are discussed. As pointed out by the reviewer, the reference to the Missoula floods were intended to just be an example of megaflooding rather than an analogy for subglacial floods beneath Antarctica. Accordingly, we have clarified that these floods were sub-aerial and sourced from a proglacial, rather than subglacial, lake (line 73).

L83: I agree here that it is inconsistent with a single/ few drainage events. But if given enough time could these not eventually produce significant channels?

We now consider this possibility – see our response to general comment 2.

L135: In the example in Figure 2a the channel illustrated is a composite feature (and also shown in Figure 6e). How would this effect the b-index? I think some discussion of this would be appropriate in the methods; it would also be useful to describe how common this configuration is in the results. Composite channel forms tend to have b-values between 1 and 2 but their exact b-value varies depending on the extent of downward incision compared to the erosion by ice at the sides of the channel. For example, if incision by meltwater is dominant then a channel's b-value will be closer to 1

(reflecting its mostly V shaped geometry) and vice versa for more U-shaped channels. This is now stated in lines 144-145. A composite configuration is reasonably common, especially for the larger channels as might be expected as these are likely older so are more likely to have been affected by ice sculpting. The enlargement of channels by ice to produce composite features is now discussed in line 395.

L253: Refer to Figure 4.

Amended.

L255: The 19 flat-bottomed depressions are not mapped in Figure 3.

We have now added the mapped basins to Fig. 3.

L257: The channels also appear to run through the basins.

Channel impressions can indeed be seen in some of the basins, and this is now acknowledged in lines 259-260: "The depressions resemble a series of basins connected, and sometimes cross cut, by the channels".

L265: Need to refer to Figure 4.

Amended.

L274-277: I like this, but am then surprised that this difference is not apparent in the form ratio (depth: width). Is there an explanation for this?

This difference is actually apparent in the form ratios; however, this was not clearly stated in the previous version of the manuscript. Although the form ratios of both regions are similar ($\sim$0.1-0.12 on average), the Labyrinth channels tend to have slightly higher form ratios than the channels in Pine Island Bay (0.08-0.2 compared to 0.04-0.14, respectively), as evidenced by the slightly less skewed distribution of form ratios for the Labyrinth compared to those offshore of Pine Island and Thwaites glaciers (Figure 4). Consequently, the Labyrinth channels tend to be deeper compared to their width (5-12 times as wide as they are deep) than those in Pine Island Bay (7-25 times

as wide as they are deep). This supports the statement in lines 275-278.

In the revised version of the manuscript, this is made clearer through the lines 276-278: "Channel cross section depth to-width ratios are also comparable in both regions, with the Labyrinth channels typically 5–12 times as wide as they are deep, whilst the channels in Pine Island Bay are slightly wider in relation to their depth at 7–25 times as wide as they are deep (Fig. 4)." and in lines 388-390: "The absence of post-incisional reworking of the top of the channels by wet based ice may explain the tendency for the Labyrinth channels to be less wide in relation to their depth than the channels in Pine Island Bay (Fig. 4)".

L345-346: Or could it be a composite meltwater signature formed over a long time. You seem to pre-empt your discussion here, and I suggest initially presenting every possible scenario.

We now consider this possibility – see our response to general comment 2.

L366-369: An alternative explanation for the Figure 6e cross-profile is it is a U-shaped channel cut by ice that has then had meltwater erosion at the base cutting the two smaller v-shaped channels. Can this scenario be ruled out?

We have added an additional figure (7) to the manuscript which provides cross sectional profiles of some of the largest channels in Pine Island Bay. The cross profiles in this figure demonstrate that not all of the large channels have the characteristic U shaped shape with V shaped incisions into the base of the channel. If all of the channels had been cut by ice prior to being eroded by meltwater, we would expect all channels to be U shaped with V shaped incisions into the base. Whilst this is the case for some of the channel cross sections in Figure 7 (e.g. Figure 7e), many of the channels are just V shaped (e.g. Figures 7b, 7c and 7d). The lack of U shaping in the upper portion of many of these channels, particularly those closest to the predicted subglacial lake basins, demonstrates that meltwater incision is the dominant process responsible for the shape of the channels and widening is a secondary process. This conclusion is

also supported by the tendency for the majority of channels to be V-shaped rather than U shaped (Figure 4).

L377: I don't follow the logic of this sentence with respect to the last – if there is enough water and a crack to initiate hydrofracture water will get to the bed.

This is a good point and we have now revised this section (lines 401-416) to include references that (i) surface melting occurs on ice shelves continent wide rather than in just the peninsula and low latitude east Antarctic ice shelves (Tedesco, 2009; Tedesco and Monaghan, 2009) and (ii) that surface meltwater can enter and be stored in the englacial system (Lenaerts et al., 2016). However, as the channels observed in Pine Island Bay have undulating long-profiles, they must have been formed beneath grounded ice and not from water that propagated through an ice shelf. Therefore, the main point of this paragraph (that under late Pleistocene full glacial conditions, the subglacial meltwater responsible for bedrock-channel formation could only have been generated by geothermal and strain heating at the ice-sheet bed) still stands and consequently remains unchanged.

L415: "with observed discharges of..."

Amended.

L428: I would not include the Bretz references here as you are referring to subglacial channel formation. I also suggest you remove the British-Irish Ice Sheet references as there is a scale difference. Some of the classic literature on bedrock cut tunnel valleys could also be referred to here (e.g. Boyd et al., 1988; Mullins and Hinchey, 1989; Kor et al., 1991; Brennand and Shaw).

As suggested, we have removed the Bretz and the British Irish Ice Sheet references here and have cited the recommended references.

L437: Can you quantify the erosion rate?

The sentence now reads: "Exceptionally high erosion rates have been associated with

supraglacial water input to the Greenland subglacial hydrological system ($4.8 \pm 2.6$ mm a-1; 1–2 orders of magnitude larger than long term estimates of denudation rates beneath the Greenland Ice Sheet) (Andrews et al., 1994; Cowton et al., 2012)".

L464: Could also cite: Livingstone, S.J., Chu, W., Ely, J.C. and Kingslake, J., 2017. Paleofluvial and subglacial channel networks beneath Humboldt Glacier, Greenland. Geology, 45(6), pp.551-554. We have now cited this reference.

L466: See also the above reference and Cooper, M.A., Michaelides, K., Siegert, M.J. and Bamber, J.L., 2016. Paleofluvial landscape inheritance for Jakobshavn Isbræ catchment, Greenland. Geophysical Research Letters, 43(12), pp.6350-6357.

We have now cited this reference.

L469: See the Livingstone et al. (2017) reference for an example of subglacial channels beneath the Greenland Ice Sheet. I also wonder whether this result is a function of the different setting of Greenland – for instance, infilling of valleys/fjords with sediments, potentially burying valleys; and less (and less detailed) sea-floor mapping.

We have added the Livingstone et al. (2017) reference to the manuscript as an example of channels beneath the contemporary Greenland Ice Sheet. We agree that less detailed sea floor mapping might affect our impression of channel abundance on the formerly glaciated Greenland continental shelf. Where high resolution multibeam echosounder mapping is available, small bedrock channels are present (e.g. Freire et al., 2015; Slabon et al., 2018); however, these channels are nothing like the large anastomosing channels observed around Antarctica in terms of their size or abundance. One possible reason for this is that the Antarctic channels are likely to be substantially older than those around Greenland so would have had more time to develop. Further mapping and morphological work around Greenland is needed to truly test this hypothesis, however. Ongoing, but as of yet unpublished, work from colleagues working in Petermann fjord (north Greenland) has also shown that there are no channelised features buried beneath the seafloor using sub-bottom profiler data to examine features

beneath the fjord sediment infill. Overall, whilst the comparison to channels in Greenland is interesting, we believe that due to a lack of data coverage, there is currently too little information to make any further statements in the manuscript than the comparison that is already present (lines 496-508). We are also wary of a discussion about the possible reasons that Greenland does not exhibit similar channels to Antarctica becoming too tangential to the overall narrative of the discussion, so we have not chosen to substantially alter that paragraph of the discussion.

[Freire, F., Gyllencreutz, R., Greenwood, S.L., Mayer, L., Egilsson, A., Thorsteinsson, T. and Jakobsson, M., 2015. High resolution mapping of offshore and onshore glaciogenic features in metamorphic bedrock terrain, Melville Bay, northwestern Greenland. Geomorphology, 250, pp.29-40.]

[Slabon, P., Dorschel, B., Jokat, W. and Freire, F., 2018. Bedrock morphology reveals drainage network in northeast Baffin Bay. Geomorphology, 303, pp.133-145.]

L486-503: How do you reconcile this paragraph with that on lines 512-523, where you cite multiple papers presenting palaeo-evidence for active subglacial lakes and drainages across bedrock?

The difference in scale observed in active lake discharges compared to the flows that the Pine Island Bay channels could accommodate was intended to emphasise the size of the channels in Pine Island Bay rather than completely exclude the role of contemporary subglacial lake drainage in the formation of the bedrock channels. The sentences that indicated this (lines 506-510 in the previous version of the manuscript) have now been removed. The association of many contemporary active subglacial lakes with soft sediments, compared to the bedrock channels in Pine Island Bay is still problematic for the use of present day subglacial lake drainage as an analogy for the mechanism which incised channels into hard crystalline bedrock in Pine Island Bay, however. This is now explained in lines 546-551. We then go on to consider lower magnitude discharges repeated over multiple glacial cycles as a possible driver

of channel incision based on the Beaud et al. (2018) modelling simulations (see lines 579-596), which allows the differences between contemporary active subglacial lake drainage events and the larger flooding events interpreted from the geological record to be reconciled.

L522: "drained" might be a better word than "utilised" here.

Amended.

Figures

Figure 3 – You also state that you identify 19 former lake basins. It would be useful to include these here if mapped? Or are these based on the modelling results?

The mapped basins were previously shown in Fig. 5d, however we have now moved these to Fig. 3. The basins now shown in Fig. 5d are based on the modelling results, so a comparison between Fig. 3 and Fig. 5 can now be used to assess how the modelling results line up with the geomorphology of Pine Island Bay.

Figure 4 – It is not clear what the dotted lines and the arrows in the V/U shaped plots refer too.

We have added the following sentence to the figure description to clarify this: "The dotted lines in the b-value plots correspond to where idealised V shaped (b = 1) and U shaped (b = 2) cross sections would fall on the histogram."

Figure 5 – Plot (d) rather reproduces Figure 3a. Could you have included the basins in Fig. 3 and rather overlay the modelled basins and channels here so the reader can directly compare how they match up. For (b) and (c) I would find it helpful if the current grounding line was included to help give some context to the reader.

We have followed your suggestion and have moved the mapped basin locations from Fig. 5d to Fig. 3, replacing them with the modelled subglacial lake locations from Fig. 5c. We have added the current ice margin into (b), (c) and (d) to give context to the

reader, as well as labelling the locations of Pine Island Glacier (PIG in Fig. 5) and Thwaites Glacier (TG in Fig. 5).

Reviewer 2: Calvin Shackleton

General comments:

This manuscript presents new mapping and morphometric analysis of subglacial channels and basins on the seafloor exposed by the retreating Pine Island and Thwaites glaciers, West Antarctica. The work also utilises a modelled LGM ice surface and isostatically corrected bed topography to model past water flow, with particular focus on water production and storage. The methodology for morphometric analysis of channels is thorough and well-explained, and the modelling approach is justified appropriately. It should perhaps be noted in the methods that the model approach does not allow for the prediction of anastomosing channels, and should be/is used only to predict water flow direction rather than simulate the behaviour of individual channels.

The results section is concise, and provides select relevant metrics from what was undoubtedly a large dataset. The figures are useful and informative, although a detailed map of the interpreted subglacial basins is currently lacking and could be incorporated into Figure 3, along with some long- and crossprofiles of the basins. This manuscript is very well written and addresses an important topic in subglacial hydrology, with an interesting discussion of the origin and cyclic behaviour of high magnitude subglacial lake drainage events and their impacts on subglacial hydrology and landscape development. I think this work should be published and I propose a small number of minor suggestions and corrections to improve the manuscript.

We have added the mapped subglacial basins into Figure 3. Comparison between figures 3a and 5d thus permit a comparison to be made between the mapped and modelled subglacial lake basins. We have also produced a new figure (7) which examines the morphology of a channel-basin system in detail and includes cross sectional profiles of both Pine Island Bay and Labyrinth channels as well as a basin. In lines

221-222 of the methods section, we now acknowledge that: "The algorithm thus calculates the volume and direction of subglacial water flow; however, it is not capable of predicting physical flow conditions within individual channels".

Specific comments:

L247: I like this interpretation, but it would be nice to see some evidence of the "lines of geological weakness" for comparison to the channels.

Not all channels are structurally controlled and we have now clarified in the revised manuscript (line 250). Examples of some channels which do appear to have a possible structural control due to a geological weakness in the bedrock are now pointed out and labelled in figure 7.

L255-259: Refer to the appropriate figure that you based your descriptions and interpretations on. At the moment I can't find a figure where the basins are clearly mapped, and suggest that a detailed description of the basins should be included, supported by select long- and cross profiles that could be incorporated into figure 3.

We have added the mapped basins into figure 3. Displaying long and cross profiles of the basins in figure 3 would make the figure too crowded in our opinion so we have chosen to produce a new figure (figure 7) which provides cross sections of a basin and channels feeding into it. This figure also compares some of the largest Labyrinth channel cross sections to channels in Pine Island Bay to demonstrate the substantially larger dimensions of the latter channels.

L344-347: This could be true, or they could have been formed over a longer period, potentially over multiple glacial cycles. Is there any other evidence that can be presented here that leads you to favour formation by large volumes of subglacial water? Or to rule out formation over longer periods? We now discuss this possibility in lines 529-596 in light of a recent paper by Beaud et al. (2018) which demonstrated that relatively low discharges can lead to significant bedrock channel erosion over the duration

of a glacial cycle. Also see our response to general comment 2 from reviewer 1.

L347-354: What if the channels were widened (and/or deepened) by sliding ice following infilling/ channel closure during their inactive phase (i.e. seasonally) rather than during subsequent ice advance/retreat?

The Antarctic setting of the channels means that widening of the channels by ice is unlikely to be a seasonal process; however, we currently do not have a good handle on the timescales over which the channels may have been widened by ice. We therefore cannot rule out shorter timescales. This is now stated in lines 376-377 which read: "Constraining the age of the submarine channels, and determining whether any resculpting by ice occurs within a glacial cycle or over shorter timescales, is difficult." We are working to investigate this issue further in future numerical modelling work, but this is beyond the scope of this publication at present.

L375: This interpretation needs to be justified a little better. Why would surface water not reach the bed? Are there no crevasses in these regions? Can you rule out hydrofracture? To me the documentation of surface meltwater rivers by Bell and Kingslake conversely indicates a high potential for surface meltwater entering the englacial and basal system.

We agree that this point requires further justification. In the present day, surface melting occurs on ice shelves continent wide but is almost non-existent in the ice sheet interior (e.g. Tedesco, 2009; Tedesco and Monaghan, 2009). For some of these ice shelves, surface meltwater has been observed to enter the englacial system and be stored as shallow sub surface lakes (Lenaerts et al., 2016). We have amended the paragraph accordingly to acknowledge this. However, as the channels have undulating long-profiles, they must have been formed beneath grounded ice and not from water that propagated through an ice shelf. Consequently, the main point of this paragraph (that under late Pleistocene full glacial conditions similar or colder than present, the subglacial meltwater responsible for bedrock-channel formation could only have been

generated by geothermal and strain heating at the ice-sheet bed) still stands and remains unchanged.

L494: Insert comma after "Thwaites Glacier"

Inserted

L519: Given that this paragraph is discussing floods from subglacial lakes, the comparison to proglacial lake Missoula seems a little out-of-place. It should be made clear that this is a proglacial lake and the comparison you are making is between their high-discharge rather than drainage environment. We have now made clear that the Missoula floods were sourced from a proglacial, rather than subglacial, lake and that this comparison is made on the basis of discharge rather than drainage type (lines 561-564). The sentence now reads: "Although the Missoula floods were released from a proglacial lake, rather than subglacial source, their estimated volume is similar to the amount of water that could have drained from beneath former Pine Island and Thwaites glaciers if a cascade of upstream lake drainage was to occur (Table 2)."

L879: "Bindschadler" missing an r

Corrected

L1227: Indicate that the long profiles are from several/select channels in the Labyrinth and Pine Island Bay region.

The figure caption now reads: "Selected long profiles of the channels comprising the Labyrinth…"

Figures:

Figure 3: I cannot see an inset map as described in the caption. Are you referring to the labelling in figure 1? It is a little confusing to have the labels on the downstream end in 3c and upstream end in 3d, I suggest to have labels only at downstream end to match the profiles in 3e and 3f below.

The 'inset map' described in the figure caption was a mistake from an earlier version of the manuscript which was subsequently replaced by Figure 1. We have removed this sentence from the caption of Figure 3 to correct this. We have followed your suggestion and moved the labels in 3d to the upstream end of the lines.

Figure 5: Perhaps include a grid on 5a to help orient readers who are not used to looking at projections of Antarctica. Here it would be nice to show the overlap between modelled basins and geomorphologically mapped basins (which I think should also be presented in figure 3).

We have added a grid on 5a to help orientate readers. We have moved the mapped basins to Fig. 3 and have replaced them with the modelled basins from Fig. 5c. Comparison between Fig. 3 and Fig. 5 can now be used to assess overlap between the mapped and modelled subglacial lake basins.

Please also note the supplement to this comment:
https://www.the-cryosphere-discuss.net/tc-2019-67/tc-2019-67-AC1-supplement.zip
* * *
Water depth (m)

0 1000 2000 3000 4000 5000 6000 7000

**Fig. 1.** Figure 1. Overview map displaying the location of features and regions referred to in the text.

[Figure]

**Fig. 2.** Figure 2. Methods of quantifying channel morphometry.

[Figure]

**Fig. 3.** Figure 3. Channelised bathymetry of the region offshore of Pine Island and Thwaites glaciers.

[Figure]

**Labyrinth**

**Pine Island / Thwaites glaciers**

**Fig. 4.** Figure 4. Size-frequency distributions of the morphometric characteristics of the channels comprising the Labyrinth and those present offshore of Pine Island and Thwaites glaciers.

**Fig. 5.** Figure 5. Modelled water flow beneath Pine Island and Thwaites glaciers at the LGM.

**(a)**

106°W

102°W

74°S

Cross sectional area (m²)
- <5000
- 10000
- 15000
- 20000
- 25000
- 30000
- 35000
- 40000
- 45000
- >50000

0   25   50
km

Fig. 7

Pine Island-Thwaites
Ice Stream margin

Modelled
subglacial lake

75°S

108°W

TG

104°W

PIG

**(b)**

Cross sectional area (m²)

8   ×10⁵

○ Distance from
mapped basins

+ Distance from
modelled basins

Decrease in
channel cross-
sectional area
with distance
from basins

0        50        100
Distance to basin (km)

**(c)**

Discharge (m³ s⁻¹)

150

100

50

0

<5000   5000-10000   10000-15000   15000-20000   >20000
Cross sectional area (m²)

**Fig. 6.** Figure 6. Spatial variation of channel size within Pine Island Bay.

[Figure]

**Fig. 7.** Figure 7. Relationship between modelled water flux and channel size through a channel basin system.

**(a)**

**Channel occupied by water**

ICE SURFACE

Ice flow

BEDROCK

(d)

Water trapped in an overdeepening

Channel incision

**(b)**

**Channel widened by ice**

ICE SURFACE

Ice flow

BEDROCK

Repetition over several glacial/ interglacial cycles

Ice widens the channel sides

**(c)**

**Present-day ice margin**

ICE SURFACE

Ice flow

BEDROCK

OCEAN WATER

(e)

Submarine channel exposed on the seafloor

**Channel after recent subglacial water incision**

**(d)**

V-shaped channel cross section

VE = 6×

Depth (m): 850 — 900 — 950
Distance (m): 0   200   400   600

**Channel after ice widening**

**(e)**

Upper channel widened by ice ←  →

Most recent period of incision

VE = 4.8×

Depth (m): 1100 — 1200 — 1300
Distance (m): 0   400   800   1200

**Fig. 8.** Figure 8. Schematic of channel formation over multiple glacial interglacial cycles.

---

## Author Response (AR1)

Comments to the Editor:

Dear Chris,

Thank you for your editorial corrections. We are happy at your recommendation. We have addressed your additional comments as follows:

Line 281: comma after "depth"?

*Added.*

Line 307: upper case 'S' in "stream"

*Corrected.*

Line 426: perhaps split the sentence, i.e. "Thus, the hydrological conditions..."

*We have now split this sentence into two:*

*"The undulating long-profiles of the channels (Figure 3f) indicate that they were formed beneath grounded ice and not from water that propagated through an ice shelf (Nitsche et al., 2013). Thus, the hydrological conditions characterising the grounded interior of the contemporary ice sheet offers the best analogue for former subglacial water generation."*

Line 579: this new sentences requires citation(s)

*The sentence now reads: "However, past outburst flooding from subglacial lakes have been suggested to have had a substantial impact on the landscape elsewhere in Antarctica (e.g. Domack et al., 2006; Lewis et al., 2006; Jordan et al., 2010; Larter et al., 2019)."*

Line 609: delete "even"?

*Deleted.*

Line 610-613: Rather long and awkward sentence. Please re-write.

[revised manuscript text omitted]